# Bioenergetics of pollen tube growth in *Arabidopsis thaliana* revealed by ratiometric genetically encoded biosensors

Jinhong Liu [1,4], Shey-Li Lim [1,4], Jia Yi Zhong [1] & Boon Leong Lim [1,2,3] ✉

Pollen tube is the fastest-growing plant cell. Its polarized growth process consumes a tremendous amount of energy, which involves coordinated energy fluxes between plastids, the cytosol, and mitochondria. However, how the pollen tube obtains energy and what the biological roles of pollen plastids are in this process remain obscure. To investigate this energy-demanding process, we developed second-generation ratiometric biosensors for pyridine nucleotides which are pH insensitive between pH 7.0 to pH 8.5. By monitoring dynamic changes in ATP and NADPH concentrations and the NADH/NAD$^+$ ratio at the subcellular level in Arabidopsis (*Arabidopsis thaliana*) pollen tubes, we delineate the energy metabolism that underpins pollen tube growth and illustrate how pollen plastids obtain ATP, NADPH, NADH, and acetyl-CoA for fatty acid biosynthesis. We also show that fermentation and pyruvate dehydrogenase bypass are not essential for pollen tube growth in Arabidopsis, in contrast to other plant species like tobacco and lily.

After landing on the stigma of a flower, a pollen grain germinates and rapidly grows a pollen tube through the style to deliver sperm to the ovule. The rapid polarized growth of the pollen tube involves functional coordination across multiple organelles and makes pollen the fastest-growing plant cell. Biosynthesis of a new plasma membrane to support pollen tube growth happens in the endoplasmic reticulum (ER), but the building blocks of this membrane, fatty acids (FAs), are produced in pollen plastids. The first step of FA biosynthesis is the conversion of acetyl-CoA into malonyl-CoA, catalyzed by acetyl-CoA carboxylase[1]. This ATP-consuming reaction is the rate-limiting step in FA biosynthesis in most organisms. Malonyl-CoA is then fed into the FA synthase complex, which comprises seven different polypeptides in plants and *Escherichia coli*, as a substrate to elongate the fatty acyl chain by two carbons in a four-step process. The four successive reactions are the condensation of activated acetyl and malonyl groups to form acetoacetyl-ACP by β-ketoacyl-ACP synthase (KAS); reduction of the carbonyl group by β-ketoacyl-ACP reductase (KAR); dehydration by β-hydroxyacyl-ACP dehydratase (HAD); and reduction of the double bond by enoyl-ACP reductase (ENR)[2]. KARs from spinach (*Spinacia*

*oleracea*) and cigar flower (*Cuphea lanceolata*) utilize NADPH as the reductant[3,4], and Arabidopsis (*Arabidopsis thaliana*), spinach, sunflower (*Helianthus annuus*), and rapeseed (*Brassica napus*) ENRs use NADH as their dedicated reductant[4-7]. Thus, FA biosynthesis requires ATP, NADPH, NADH, and acetyl-CoA in Arabidopsis plastids.

There are two possible sources of plastid ATP, either via plastid glycolysis or by the import of cytosolic ATP through nucleotide transport proteins (NTTs)[8]. Likewise, there are two possible sources of NADPH in non-photosynthetic plastids: the oxidative pentose phosphate pathway (OPPP) and NADP-malic enzyme (NADP-ME). Arabidopsis has only one plastid malic enzyme, NADP-ME4, whose encoding gene is expressed in multiple tissues, including pollen[9]. This enzyme can convert malate and NADP$^+$ into pyruvate and NADPH. The building block of FA, acetyl-CoA, can be generated from pyruvate or acetate via catalysis by plastid pyruvate dehydrogenase (pPDH)[10] or acetyl-CoA synthetase (ACS)[11] of the PDH bypass[12], respectively. pPDH converts pyruvate into acetyl-CoA and reduces NAD$^+$ into NADH. Plastid pyruvate itself can originate from three possible sources: plastid glycolysis, NADP-ME4[9], or the import of cytosolic pyruvate through the

[1]School of Biological Sciences, University of Hong Kong, Hong Kong, China. [2]HKU Shenzhen Institute of Research and Innovation, Shenzhen, China. [3]State Key Laboratory of Agrobiotechnology, The Chinese University of Hong Kong, Hong Kong, China. [4]These authors contributed equally: Jinhong Liu, Shey-Li Lim. ✉e-mail: bllim@hku.hk

pyruvate transporter BILE ACID:SODIUM SYMPORTER FAMILY PRO-TEIN2 (BASS2) located on the chloroplast membrane[13,14]. After incubation of tobacco (*Nicotiana tabacum*) pollen with [14]C-labeled ethanol, [14]C label can be detected in $CO_2$ and lipids, supporting a role for the PDH bypass in lipid biosynthesis during tobacco pollen tube growth[15]. In the Arabidopsis genome, there are 4 pyruvate decarboxylase (PDC) genes[16], 1 alcohol dehydrogenase (ADH) gene[17], 16 aldehyde dehydrogenase (ALDH)[18], and 1 ACS gene[11]. While all four PDCs are located in the cytosol, the single ADH localizes to the cytosol and the nucleus[19], and the single ACS is a plastid protein[11]. Of the 16 ALDH proteins encoded by the Arabidopsis genome[18], ALDH3I1[20] and ALDH10A8[21] are targeted to chloroplasts and leucoplasts, respectively. To date, only family 2 ALDH members (ALDH2B4, ALDH2B7, and ALDH2C4) have been shown to oxidize acetaldehyde to acetate[18,22]. There are also five possible sources of NADH in non-photosynthetic plastids: glyceraldehyde-3-phosphate dehydrogenase C (GAPCp) of plastid glycolytic pathway[23], pPDH, plastid NAD-malate dehydrogenase (pNAD-MDH)[7], plastid ALDH[24], and plastid 3-phosphoglycerate dehydrogenase (pPGDH)[25].

Here, to delineate the above processes, we introduced *in planta* ATP[26], NADPH, and NADH/NAD+ ratiometric biosensors[27,28] into Arabidopsis pollen. Since the first-generation NADPH and NADH/NAD+ ratiometric biosensors are pH-sensitive, an independent transgenic line expressing a control biosensor is needed for post-experimental pH normalization[27,28]. To visualize dynamic changes of NADPH levels and NADH/NAD+ ratio in individual pollen plastids or in the cytosol without pH normalization, we developed pH insensitive (pH 7.0 to 8.5) versions of these pyridine nucleotide biosensors by fusing a mCherry fluorophore at the N termini of the iNAPs and SoNar, as a mCherry-iNap1 fusion protein has been reported previously[29]. We generated transgenic lines that expressed these second-generation biosensors in plastids and the cytosol under the control of the pollen-specific LAT52 promoter. We then employed various inhibitors and T-DNA insertion mutants or EMS mutants[7] inactivating individual enzymes from the above-mentioned biochemical pathways, to examine the biological roles of pollen plastids, the source of energy and building blocks for pollen tube membrane biosynthesis, and the underlying biochemical pathways. We propose that pollen plastids play an important role in transforming carbon and energy from exogenous sugars into FAs to achieve rapid pollen tube growth. Our ultimate goal is to establish a clearer model of the energy flow during pollen tube growth that links plastids, energy flow, and FA biosynthesis.

## Results

### Development of mCherry-SoNar/iNAPs biosensors

To overcome the pH sensitivity of the first-generation SoNar and iNAPs biosensors, a mCherry fluorophore was fused to the N termini of the original SoNar and iNAPs biosensors with a pentapeptide linker (GGSGG)4 (Fig. 1a and Table 1). iNAPs are the mutated variants of SoNar[29] and both employed a similar strategy of inserting a circular permuted yellow fluorescent protein (cpYFP) between the two subunits of the *Thermus aquaticus* Rex (T-Rex) repressors (Fig. 1a). Due to the possible differences in NAPDH levels in various subcellular compartments, two NADPH biosensors, named mCherry-iNAP1 (higher affinity) and mCherry-iNAP4 (lower affinity), were developed, respectively. To reveal the response characteristic of the mCherry-SoNar and mCherry-iNAPs (mCherry-iNAP1 and mCherry-iNAP4) excitation spectra, we scanned the absorbance spectra of purified mCherry-SoNar and mCherry-iNAPs recombinant proteins. The recombinant biosensors clearly displayed dual absorption peaks at around 405 nm and 580 nm (Fig. 1b).

To test the in vitro properties of the biosensor proteins, we first determined their binding affinity and pH sensitivity. The mCherry-SoNar and mCherry-iNAPs biosensors responded consistently under

the physiological pH range from pH 7.0 to 8.5 when they were excited at 405 nm and 580 nm, suggesting that these second-generation biosensors are pH insensitive between pH 7.0 to 8.5 as compared to the first-generation SoNar and iNAPs biosensors (Fig. 1c, d and Supplementary Fig. 1c, f, i). The $K_d$ value of mCherry-SoNar was determined as 0.04, which is similar to the first-generation SoNar biosensor (Table 1). The second-generation mCherry-iNAP1 and mCherry-iNAP4 variants showed slightly lower binding affinity to NADPH, that is, with $K_d$ values of 0.94 μM and 39 μM, respectively, compared to 0.29 μM and 30 μM for the first-generation of iNAP1 and iNAP4 biosensors at 25 °C (Table 1)[27]. Notably, because accurate determination of biosensor affinities in vivo is technically difficult, the in vitro affinities determined here can only be taken as references. The biosensors may or may not have identical dynamic ranges and affinities when they are expressed in different subcellular compartments.

Upon application of exogenous NADPH and NADH/NAD+, emission detection of mCherry-SoNar and mCherry-iNAPs changed ratiometrically, where the emission excited at 405 nm increases proportionally with the increment of pyridine nucleotide concentration, while the emission excited at 580 nm remains constant (Supplementary Fig. 1a, b, d, e, g, h). The responsiveness of the first-generation SoNar and iNAPs proteins to L-ascorbic acid, reduced glutathione, hydrogen peroxide ($H_2O_2$), menadione, and dithiothreitol (DTT) have been previously studied[27]. Similar to the first-generation biosensors, the stability of the mCherry-SoNar and mCherry-iNAPs were maintained, and we confirmed that the mCherry-SoNar and mCherry-iNAPs proteins were not affected by exogenous $H_2O_2$, menadione, L-ascorbic acid, and reduced glutathione, but they remained sensitive to DTT (Fig. 1e). To examine if exogenously supplied inhibitors could potentially affect biosensor protein structures, we examined the effects of commonly used plant inhibitors named rotenone (inhibits Complex I), thenoyltrifluoroacetone (TTFA) (inhibits Complex II), antimycin A (inhibits Complex III), oligomycin (inhibits ATP synthase), potassium cyanide (KCN) (inhibits cytochrome c-dependent respiration), 1-aminoethylphosphonic acid (AEP) (inhibits pyruvate dehydrogenase complex), salicylhydroxamic acid (SHAM) (inhibits alternative oxidase), disulfiram (inhibits aldehyde dehydrogenase), CGP3466B maleate, and iodoacetate (inhibits glyceraldehyde-3-phosphate dehydrogenase) on purified mCherry-SoNar and mCherry-iNAPs proteins. None of these chemical inhibitors affect the biosensor ratios of purified mCherry-SoNar and mCherry-iNAPs proteins (Supplementary Fig. 2).

### Generation of transgenic Arabidopsis expressing biosensors in the plastids and cytosol of Arabidopsis pollen

We then stably expressed the At1.03, mCherry-SoNar, and mCherry-iNAPs biosensors, under the control of the LAT52 pollen promoter of *Solanum lycopersicum*, in the cytosol and plastids of Arabidopsis pollen tubes (Supplementary Fig. 3). For each biosensor and compartment, we obtained at least three independent transgenic lines. Pollen tube samples were prepared in confocal dishes and glass slides before being examined with a confocal microscope. The pseudocolor hue, saturation, lightness (HSV) ratio images, and quantitative comparison of the independent lines for each biosensor and compartment revealed no significant differences, reflecting their similar behavior (Supplementary Fig. 3). Therefore, we selected a single line per biosensor and compartment for subsequent experiments. We detected no significant differences in pollen tube length, width, or germination rates in the homozygous biosensor lines compared to the WT (Supplementary Fig. 4), indicating that the expression of these biosensors has no effect on pollen physiology or morphology.

To confirm that the biosensors fused to the transketolase chloroplast target peptide (TKTP)[30] correctly localized to pollen plastids, we generated two pollen plastid marker lines expressing *FtsZ1-mRFP* (encoding a fusion protein between FtZ1 and monomeric red

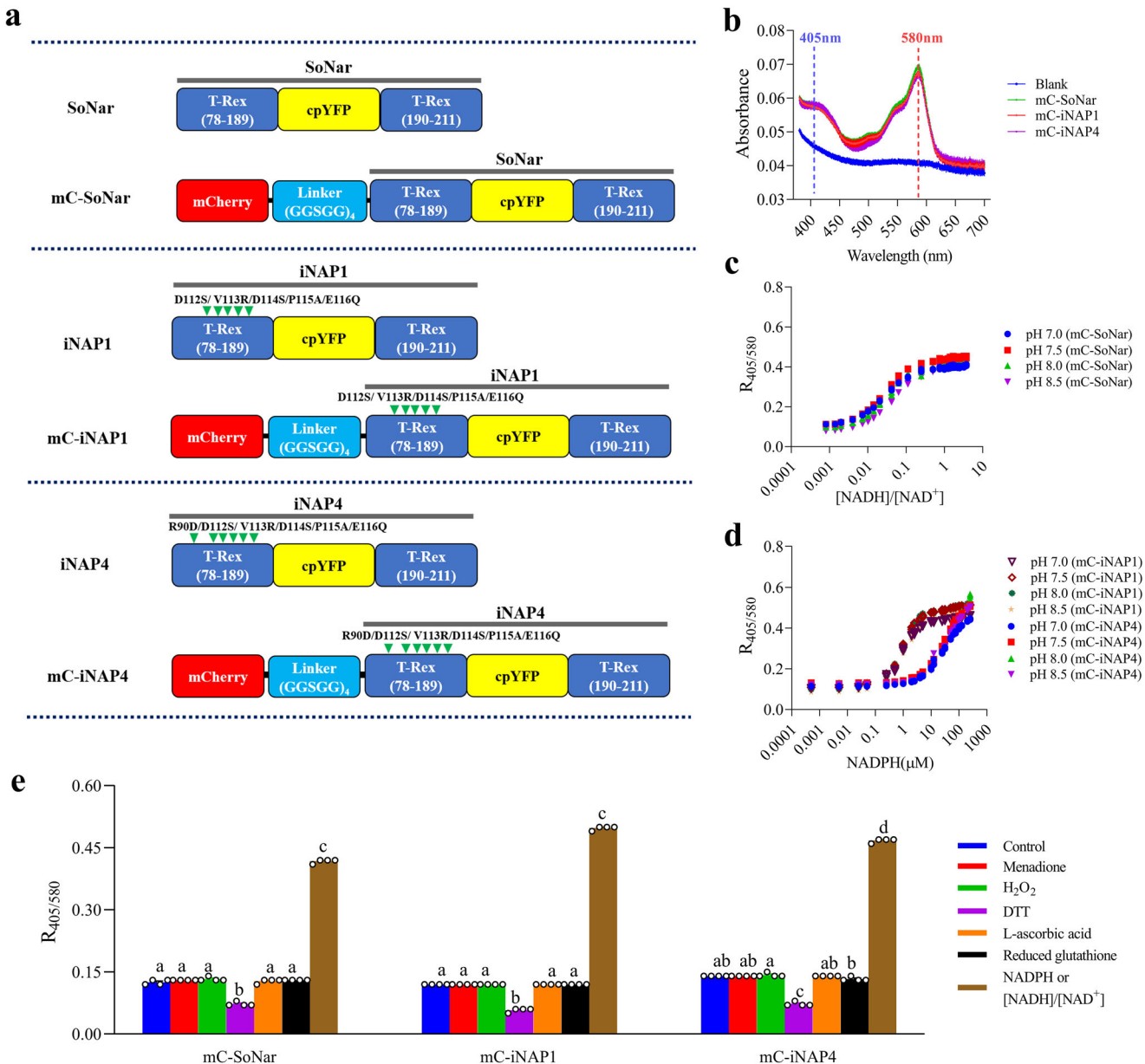

**Fig. 1 | Characterization of mCherry-SoNar and mCherry-iNAP1 and iNAP4.**
**a** Schematic diagrams of the structure of the SoNar, iNAPs, mCherry-SoNar, and mCherry-iNAPs biosensors. Arrowheads in green indicate mutations in natural ligand-sensing domains. There are five mutations on T-Rex (D112S, V113R, D114S, P115A, and E116Q) on iNAP1 and mCherry-iNAP1[29], and six mutations on T-Rex (R90D, D112S, V113R, D114S, P115A, and E116Q) on iNAP4 and mCherry-iNAP4[29]. **b** Absorbance spectra of recombinant mCherry-SoNar, mCherry-iNAP1 and mCherry-iNAP4 ($n = 3$ biologically independent samples; error bars ± SEM). **c** NADH/NAD+ titration curve of recombinant mCherry-SoNar ($n = 4$ biologically independent samples; error bars ± SEM). **d** NADPH titration curve of recombinant mCherry-iNAP1 and mCherry-iNAP4 ($n = 4$ biologically independent samples; error

bars ± SEM). **e** In vitro ratios of recombinant mCherry-SoNar, mCherry-iNAP1, and mCherry-iNAP4 treated with different oxidants (15 μM menadione or 0.5 mM $H_2O_2$), reductant (5 mM dithiothreitol (DTT)), antioxidants (5 mM L-ascorbic acid or 5 mM reduced glutathione), and 250 μM NADPH or 50 μM [NADH]/[NAD+] at 1:1 ratio in the pseudocytosol buffer of pH 7.5. The significant differences among the treatments are indicated in different letters as determined by Tukey's HSD ($P < 0.05$); $n = 4$ biologically independent samples; error bars ± SEM. Exact $p$-values for panel e experiments are provided in the source data file. The recombinant proteins were excited at 405 nm and 580 nm in a fluorescence plate reader and the ratios of their emissions at 520 ± 16 nm and 609 ± 25 nm were shown. All the reading temperature was set as 25 °C. mC, mCherry.

fluorescent protein [mRFP]) or *FtsZ1-mCerulean* (encoding a fusion protein between FtZ1 and the monomeric cyan fluorescent protein Cerulean)[31], under the control of the LAT52 promoter. In pollen tubes of the F1 hybrids (*TKTP-At1.03 × FtsZ1-mRFP* and *TKTP-mcherry-iNAP1/iNAP4/SoNar × FtsZ1-mCerulean*), all biosensor signals exclusively colocalized with FtsZ1-associated fluorescence (Supplementary Fig. 5). Since mCherry-iNAP1/iNAP4/SoNar have wide excitation windows peaking around 405, 488, and 543 nm, respectively, we also evaluated the extent of fluorescence crosstalk from these fluorescent proteins in

the mCerulean (excited at 458 nm) detection channel (Supplementary Fig. 6). Using the same imaging settings, the fluorescence crosstalk from mCherry-iNAP1/iNAP4/SoNar (Supplementary Fig. 6b–d) in the 458 nm channel was negligible. Although we detected a little fluorescence bleed-through in the 405 nm channel when mCerulean was excited (Supplementary Fig. 6a), the fluorescence signals detected in the 458 nm channel for mCherry-iNAP1/iNAP4/SoNar and FtsZ1-mCerulean in pollen tubes of the crossed line originated from mCerulean but not from the biosensors.

**Table 1 | Properties of the first- and second-generation iNAPs and SoNar biosensors**

| Properties | iNAPs[27,29] | mC-iNAPs | SoNar[27,72] | mC-SoNar |
|---|---|---|---|---|
| Species sensed | NADPH | NADPH | NADH/NAD+ | NADH/NAD+ |
| Sensor type | Ratiometric | Ratiometric | Ratiometric | Ratiometric |
| Fluorescent protein | cpYFP | cpYFP, mCherry | cpYFP | cpYFP, mCherry |
| Ex/Em | ■ Ex 405/Em 520<br>■ Ex 488/Em 520 | ■ Ex 405/Em 520<br>■ Ex 543-580/Em 630 | ■ Ex 405/Em 520<br>■ Ex 488/Em 520 | ■ Ex 405/Em 520<br>■ Ex 543-580/Em 630 |
| $K_d$ (25 °C) | ■ iNAP1: ~ 0.29 µM<br>■ iNAP4: ~30 µM | ■ mC-iNAP1: ~0.94 µM<br>■ mC-iNAP4: ~39 µM | ■ SoNar $_{NADH/NAD^+}$: ~ 0.04 | ■ mC-SoNar $_{NADH/NAD^+}$: ~ 0.04 |
| Detection range | ■ iNAP1: 0.05 µM - 8 µM<br>■ iNAP4: 2 µM - 50 µM | ■ mC-iNAP1: 0.05 µM - 5 µM<br>■ mC-iNAP4: 2.5 µM - 100 µM | ■ SoNar $_{NADH/NAD^+}$: 0.002 -0.75 | ■ mC-SoNar $_{NADH/NAD^+}$: 0.004 – 0.5 |
| pH sensitivity | Sensitive at 488 nm | Insensitive (pH 7.0–8.5) | Sensitive at 488 nm | Insensitive (pH 7.0 – 8.5) |
| Validated *in planta* | Yes[27] | This study | Yes[27] | This study |

*Ex/Em* excitation wavelength/emission wavelength, *mC* mCherry, *cpYFP* circularly permuted yellow fluorescent protein, *mC-iNAPs* mCherry-iNAP1/4.

## Sources of ATP in the plastids of growing pollen tubes

We targeted the ATP biosensor At1.03 and its non-responsive control biosensor At1.03^R122K/R126K to plastids or the cytosol of growing pollen tubes (Fig. 2a). We determined that the ATP concentration is higher in the cytosol than in the plastid stroma (Fig. 2b, c), suggesting that plastids are not the main source of ATP in pollen. Pollen grains contain numerous mitochondria; for instance, maize (*Zea mays*) pollen contain ~20 times more mitochondria per cell than vegetative tissues[32]. The inhibitor of mitochondrial Complex I, rotenone, caused a modest but significant decrease in cytosolic and stromal ATP levels (Fig. 2d, e). Treatment with the H+-ATP-synthase inhibitor oligomycin strongly lowered both cytosolic and stromal ATP levels in pollen tubes (Fig. 2d, e) and blocked pollen germination and pollen tube growth (Fig. 2f–i). By contrast, the glycolysis inhibitors iodoacetate and CGP3466B suppressed pollen tube growth (Fig. 2f–i) without affecting cytosolic ATP levels (Fig. 2j, k). These results supported the notion that mitochondria are the major ATP supplier for the cytosol.

The import of cytosolic ATP into plastids requires NTTs. We observed that *NTT2*, but not *NTT1*, is strongly expressed in growing pollen tubes, as evidenced by their respective promoters driving the expression of the reporter gene *β*-glucuronidase (GUS) (Fig. 3a, b). However, the *ntt1* and *ntt2* single mutants did not show any defects in pollen tube growth (Fig. 3c–g) or changes in stromal ATP levels (Fig. 3h, i). We then carried out in vitro pollen tube assays on the homozygous double knockout line of *NTT1* and *NTT2* (*ntt1/2*) (Supplementary Fig. 7a, b)[8]. While its pollen germination rate was significantly lower than that of WT, for those successfully germinated, the pollen tube growth was comparable with the WT pollen (Supplementary Fig. 7c–f). Therefore, while both plastid glycolysis and the import of cytosolic ATP are sources of stromal ATP in WT pollen, plastid glycolysis alone *in ntt1/2* can supply a substantial amount of stromal ATP for pollen tube growth.

## Sources of NADPH in the plastids of growing pollen tubes

There are also two possible sources for NADPH production in non-photosynthetic plastids: the OPPP and NADP-ME. Of the four NADP-MEs in Arabidopsis, only NADP-ME4 localizes to chloroplasts, and its encoding gene is highly expressed in pollen based on promoter *GUS* reporters[9]. In a pollen tube growth assay, pollen grains of two *nadp-me4* mutants exhibited shorter pollen tubes compared to those of the WT after 4 h incubation in pollen germination medium (Fig. 4a–d). To test whether NADPH production contributed to this phenotype, we introduced the *TKTP-mCherry-iNAP4* transgene into the *nadp-me4-1* mutant. The loss of NADP-ME4 function in pollen resulted in a mild decrease in NADPH levels within plastids (Fig. 4e, f). Therefore, NADP-ME4 is required for pollen tube growth and supplies NADPH in pollen plastids.

To examine whether the OPPP also produces NADPH within pollen plastids, we treated WT pollen with the OPPP inhibitor

6-aminonicotinamide (6-AN). 6-AN did not significantly compromise pollen tube germination and growth over a wide range of concentrations (10–500 µM), although treatment with a higher dose (5 mM) led to shorter but wider pollen tubes (Fig. 4g–j). In addition, 30 min treatment with 5 mM 6-AN did not reduce plastid NADPH levels in the TKTP-mCherry-iNAP4 biosensor line in the WT background (Fig. 4e, f), further supporting the idea that NADP-ME4 is the major NADPH supplier in plastids of growing pollen tubes. However, blocking the OPPP in the *nadp-me4* mutants exacerbated its pollen tube growth defects, as the length of *nadp-me4* pollen tubes treated with 5 mM 6-AN was only 35% of untreated WT, which was much more severe than that of either untreated *nadp-me4* mutants (84% of untreated WT) or 6-AN-treated WT pollen (~57% of untreated WT) (Fig. 4). We conclude that both the OPPP and the NADP-ME pathway supply NADPH for pollen tube growth, but NADP-ME is the major source of NADPH in plastids.

## Sources of pyruvate in the plastids for the growth of pollen tubes

In pollen plastids, pyruvate can be imported from the cytosol by the translocator BASS2 or synthesized through oxidative decarboxylation of malate or during plastid glycolysis from phosphoenolpyruvate (PEP). Oxidative decarboxylation of malate by plastid NADP-ME4 is a major source of NADPH, which should also supply a substantial amount of pyruvate to plastids. To evaluate the possible contribution of the import of cytosolic pyruvate, we measured pollen tube growth in two knockout mutants for *BASS2*. We detected no significant differences in pollen tube length, width, or pollen germination rate in two putative *bass2* null mutants compared to the WT (Fig. 5a–d). There are three plastid pyruvate kinase (*PKp*) genes in the Arabidopsis genome. According to the published microarray data (Supplementary Table 1) and our reverse transcription–quantitative PCR (RT-qPCR) data (Supplementary Fig. 8), *PKp3* exhibits the highest mRNA expression level in Arabidopsis pollen tubes, suggesting that it is the major PKp that supplies pyruvate to pollen plastids during pollen tube growth. While the pollen tubes of *pkp1* and *pkp2* single mutants grew normally (Fig. 5e–h), no *pkp3* mutant with T-DNA insertion in an exon is available. Nonetheless, since plastid glycolysis supplies a substantial amount of stromal ATP, it is reasonable to conclude that both malate decarboxylation and plastid glycolysis supply pyruvate to pollen plastids.

## Sources of acetyl-CoA for FA biosynthesis

The building block of FA, acetyl-CoA, can be generated from pyruvate or acetate via catalysis by pPDH[10] or ACS[11] of the PDH bypass, respectively[12]. The PDH bypass was suggested to be more important than the PDH pathway in tobacco pollen tubes to provide acetyl-CoA for FA biosynthesis, as the ALDH inhibitor disulfiram caused stronger inhibition of pollen tube growth than the PDH

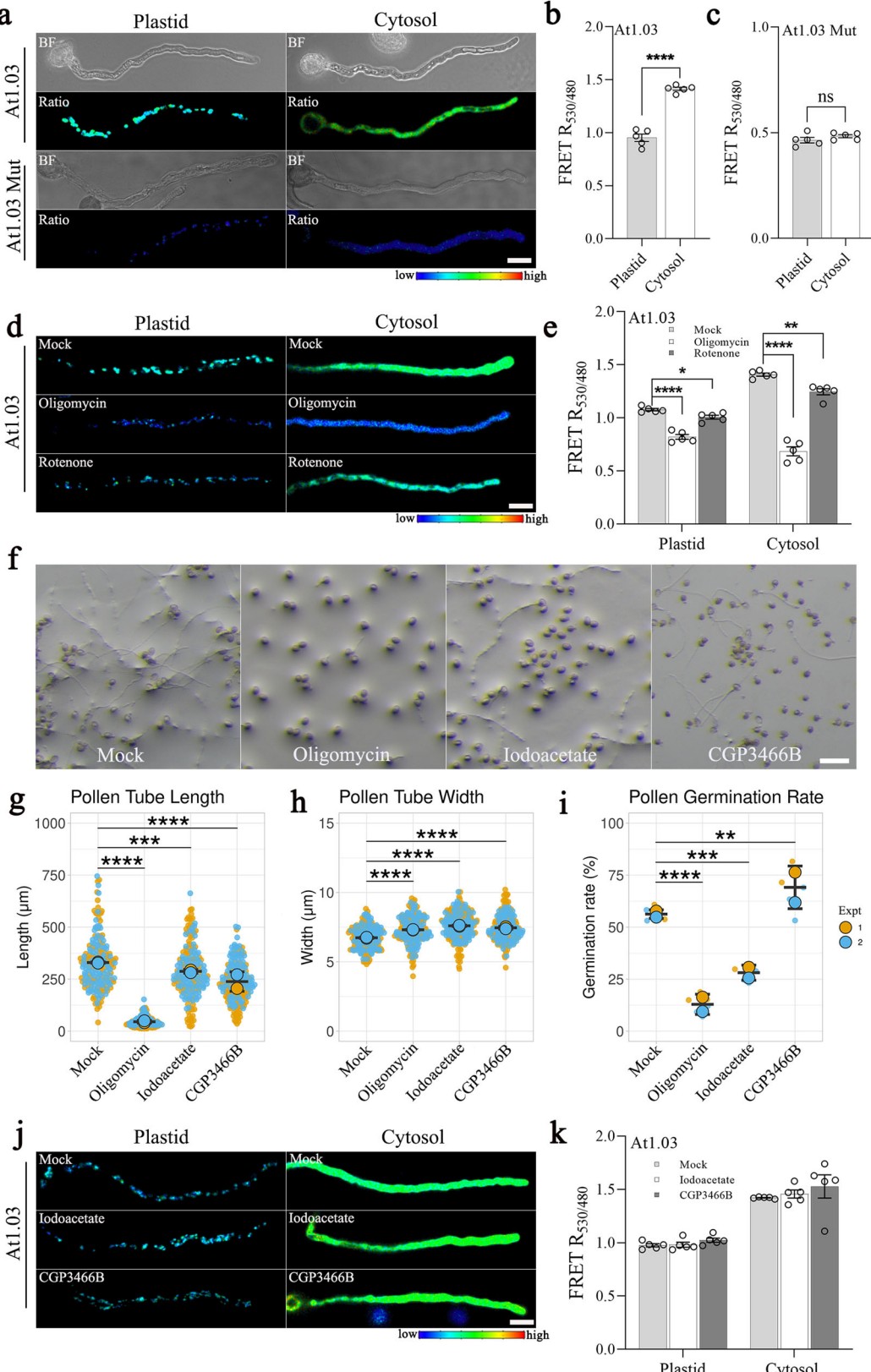

inhibitor AEP[33]. Similarly, in Arabidopsis, a 90 μM AEP treatment led to lower pollen germination rates and moderately shorter pollen tubes, while treatment with 5 μM disulfiram significantly impaired pollen germination and elongation and induced rupture of pollen tube tips (Fig. 6a–d). Therefore, based on inhibitor studies, the PDH bypass may be more important than the PDH

pathway for Arabidopsis pollen tube growth. However, we did not notice any defects during in vitro pollen germination assays of homozygous *acs*, *acn1*, or *adh1* mutants (Fig. 6e–h; Supplementary Fig. 9). Thus, the inhibition of pollen tube growth by disulfiram may be attributed to the general and nonspecific suppression of multiple ALDHs.

**Fig. 2 | Source of ATP in growing pollen tubes. a–c** The ATP biosensor At1.03 showed that the ATP concentration was lower in pollen plastids than in the cytosol, whereas the ratios of non-responsive biosensor At1.03[R122K/R126K] were equal in the two compartments. Emissions at 470–507 nm and 526–545 nm with the same excitation at 458 nm were recorded. $n = 5$ biologically independent pollen tubes in each column; error bars ± SEM, a two-sided unpaired $t$-test was used. $p$ values of panel b = 2.077 ×10$^{-6}$ and panel c = 0.244. Scale bar = 20 μm. **d, e** Effects of 40 μM oligomycin A (ATP synthase inhibitor) and 50 μM rotenone (mETC complex I inhibitor) on ATP biosensor ratios. The inhibitors were added after 4 h pollen germination on agar with 18% (w/v) sucrose, and the ratios of pollen tubes ($n = 5$ in each column; error bars ± SEM) were measured after 30 min inhibitor treatment. $p$ values of panel e = 6.551 ×10$^{-7}$ (plastid: mock vs. oligomycin), 0.031 (plastid: mock vs. rotenone), 2.184 ×10$^{-8}$ (cytosol: mock vs. oligomycin), and 5.748 ×10$^{-3}$ (cytosol: mock vs. rotenone). Asterisks indicate statistically significant differences (***$P < 0.001$, one-way ANOVA with Dunnett's multiple comparison test). **f–i** WT pollen tubes subjected to mock treatment, 5 nM oligomycin, 1 μM iodoacetate, and 40 μM CGP3466B. After 4 h germination, pollen tubes become shorter but wider when subjected to oligomycin, iodoacetate, and CGP3466B treatment. $n = 100$

pollen tubes were measured for tube length and width in each experiment (Expt). $n = 3$ groups with at least 500 pollen each were counted for the pollen germination rate in each experiment. Error bars ± S.D. of mean value of each replicate. $p$ values of panel g = 3.787 ×10$^{-8}$ (mock vs. oligomycin), 1.161 ×10$^{-4}$ (mock vs. iodoacetate), and 3.787 ×10$^{-8}$ (mock vs. CGP3466B). $p$ values of panel h = 9.733 ×10$^{-8}$ (mock vs. oligomycin), 3.787 ×10$^{-8}$ (mock vs. iodoacetate), and 3.787 ×10$^{-8}$ (mock vs. CGP3466B). $p$ values of panel i = 2.587 ×10$^{-8}$ (mock vs. oligomycin), 2.556 ×10$^{-7}$ (mock vs. iodoacetate), and 3.504 ×10$^{-3}$ (mock vs. CGP3466B). Asterisks indicate statistically significant differences (***$P < 0.001$, one-way ANOVA with Dunnett's multiple comparison test). Scale bar = 100 μm. **j, k** Effects of glycolysis inhibitors iodoacetate (100 μM) and CGP3466B (200 μM) on ATP biosensor ratios. The inhibitors were added after 4 h of pollen germination on agar with 18% (w/v) sucrose, and the images were taken after 30 min inhibitor treatment ($n = 5$ biologically independent pollen in each column; error bars ± SEM). $p$ values of panel k = 0.971 (plastid: mock vs. iodoacetate), 0.189 (plastid: mock vs. CGP3466B), 0.904 (cytosol: mock vs. iodoacetate), and 0.443 (cytosol: mock vs. CGP3466B). Asterisks indicate statistically significant differences (***$P < 0.001$, one-way ANOVA with Dunnett's multiple comparison test). BF bright field.

In the PDH bypass, acetaldehyde is converted to acetate by ALDH[34]. To examine whether this enzymatic step takes place within pollen plastids or in any other compartments, we evaluated the expression profiles of several Arabidopsis *ALDH*s in pollen tubes by RT-qPCR or GUS histochemical assays. We established that the genes encoding the only two plastid-localized ALDHs (chloroplast-localized ALDH3I1 and leucoplast-localized ALDH10A8) were not expressed in pollen tubes (Fig. 6i, j). Among other ALDHs that may participate in the acetaldehyde–acetate reaction, the genes encoding cytosolic ALDH2B7 and mitochondrial ALDH7B4 were highly expressed in growing pollen tubes, as shown by GUS staining (Fig. 6i) and RT-qPCR data (Fig. 6j and Supplementary Fig. 8). Hence, acetaldehyde is likely to be converted to acetate by ALDH2B7 in mitochondria but not in plastids. The lack of growth defects in the *acs* mutant also indicated that the PDH bypass is not essential for Arabidopsis pollen tube growth, making the plastid PDH pathway the key source of acetyl-CoA for FA biosynthesis.

### Sources of NADH in the plastids of growing pollen tubes

There are five possible sources of NADH in non-photosynthetic plastids: plastid glycolysis, pPDH, pNAD-MDH[35], plastid ALDH[24], and pPGDH[25]. The two Arabidopsis *ALDH*s encoding plastid enzymes (AtALDH3I1 and AtALDH10A8) are not expressed in pollen tubes, excluding the possibility of conversion of acetaldehyde to acetate to release NADH in plastids. Furthermore, AEP and disulfiram treatments decreased both cytosolic and plastid NADH/NAD$^+$ ratios but not NADPH levels (Fig. 7a–d). These data confirmed that the plastid PDH pathway and mitochondrial ALDH supply substantial amounts of NADH in plastids and mitochondria, respectively. pNAD-MDH is likely to generate malate and NAD$^+$ instead of NADH in plastids to support plastid glycolysis and supply malate for NADP-ME4. We determined that this step is important, as a mutant (*suppressor of mod1 410* [*som410*])[7] harboring an A90V amino acid substitution in pNAD-MDH was characterized by slightly shorter pollen tubes (Fig. 7e–h). This corroborated with the observation that 50% pollen of heterozygous *pnad-mdh* knockout mutant did not germinate[36]. pPGDH could also be a source of NADH in pollen plastids, as a substantial amount of *pPGDH1* transcript is expressed in pollen tube (Supplementary Table 1, Supplementary Fig. 8) and knocking out of pPGDH1 (EDA9) is embryo lethal[37]. The pPGDH pathway should be a major source of serine for pollen tube growth, as there is no photorespiration. The embryo lethal phenotype of the *pdgh1* mutant is likely due to the lack of serine supply, rather than the lack of NADH supply by pPGDH1. Hence, plastid glycolysis and pPDH, and possibly pPDGH, are the major sources of stromal NADH during pollen tube growth (Fig. 8).

## Discussion

While pollen starch and exogenous sucrose are generally regarded as the main source of energy for pollen tube growth[36], pollen also contain a substantial amount of triacylglycerols, which are stored within oil bodies (OBs). In olive (*Olea europaea*), pollen can germinate and pollen tubes can grow normally without exogenous sucrose in in vitro assays, while the inhibition of lipase activity by ebelactone B strongly inhibits pollen germination, supporting the idea that stored lipids in mature olive pollen can be a major energy source for initial pollen tube growth[38]. In olive, OBs in pollen disappear in the pollen tube 7 h into in vitro germination assays[39]. However, in Arabidopsis, lipids remain detectable even after initiation of pollen tube growth, as more Nile red–stained OBs can be seen throughout the growing pollen tube[38], indicating that lipids are being synthesized during pollen tube growth. Arabidopsis pollen grains contain roughly 40–50 plastids, each with a diameter of ~2 μm$^2$ that accumulate starch grains, as observed in mature non-germinated Arabidopsis pollen[31]. While such starch grains in plastids can provide the carbon skeletons and energy necessary for the initial stages of pollen germination, a continuous supply of sucrose from the style is required to sustain pollen tube growth. There are at least two cell wall invertases and four sucrose transporters present at the plasma membrane of Arabidopsis pollen[40]. Imported sucrose can be converted by cytosolic invertases into fructose and glucose[41,42], and the latter can be further converted to glucose-6-phosphate (G6P) by hexokinases[40]. The Arabidopsis genome encodes two plastid-targeted G6P/phosphate translocators, GPT1 and GPT2[43]. The importance of G6P import into plastids for pollen germination and pollen tube growth is clear in Arabidopsis, as the loss of GPT1 function in *gpt1* mutants hampers pollen germination, pollen tube growth, and OB formation[44]. Mature Arabidopsis pollen expresses *GPT1* to high levels, but the expression of the genes encoding the triose phosphate/phosphate translocator (TPT), the PEP/phosphate translocator (PPT1), and GPT2 was very low or not detectable, suggesting that only limited amounts of triose-P or PEP are imported into pollen plastids[44].

Imported G6P into plastids may serve two purposes: to generate NADPH via the OPPP and to supply carbon skeletons for plastid glycolysis and FA biosynthesis (Fig. 8). Our data demonstrate that plastid NADP-ME4, rather than the OPPP, is the major supplier of NADPH in pollen plastids (Fig. 4). Hence, the imported G6P is mainly consumed by plastid glycolysis, and the flux of plastid glycolysis should be higher than that of the OPPP. Plastid glycolysis generates ATP, NADH, and pyruvate, which are all required for FA biosynthesis. In addition to the carbon skeletons derived from G6P, FA biosynthesis in Arabidopsis also requires ATP, NADPH, and NADH[4–7]. FA biosynthesis consumes a fixed stoichiometry of these three molecules[45]. Our previous work on mesophyll chloroplasts showed that stromal ATP is rapidly consumed

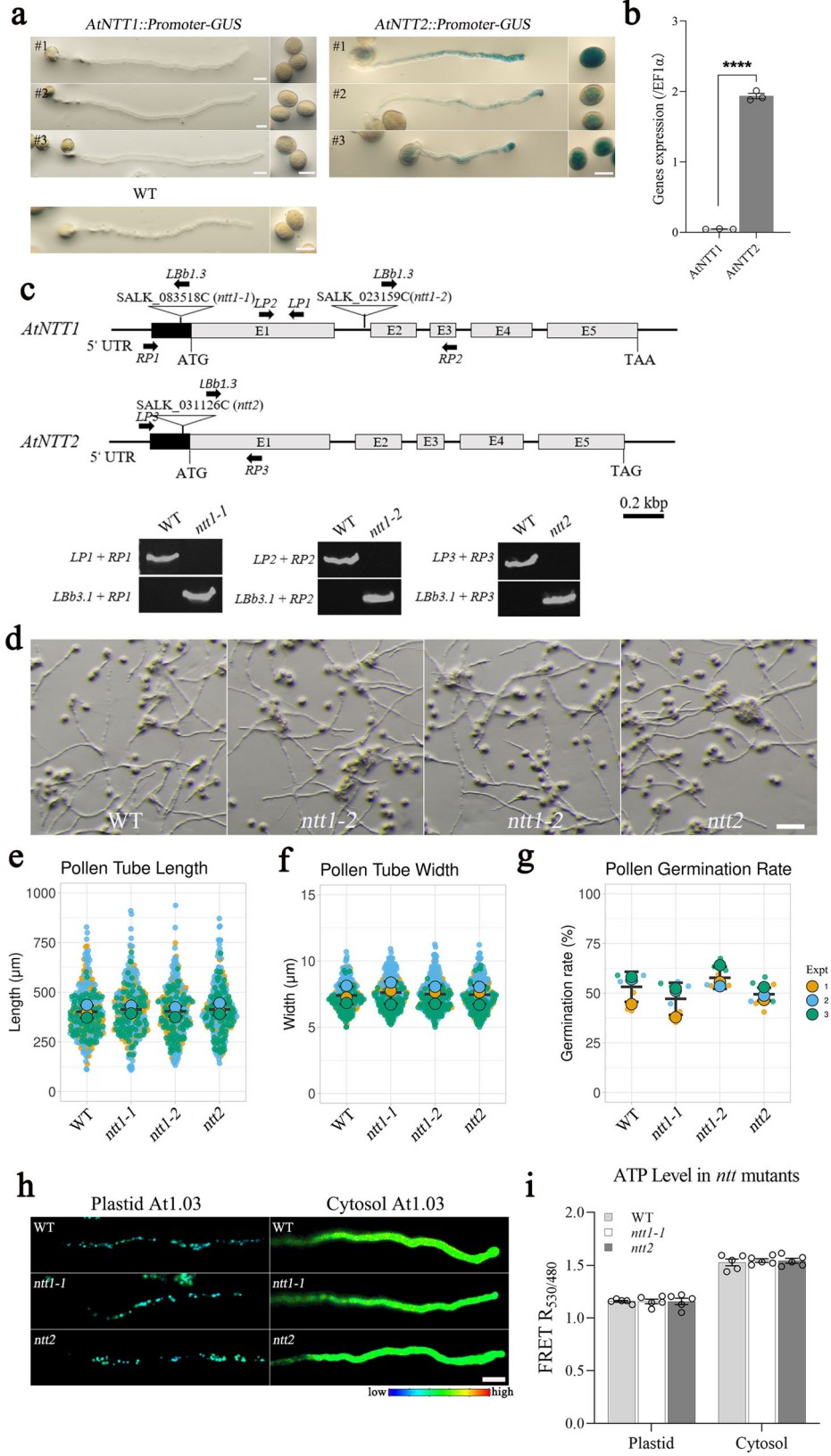

in the dark and that ATP is limiting for carbon fixation via the Calvin–Benson–Bassham cycle[26]. Surplus stromal NADPH generated from the linear electron flow is exported in the form of malate via the malate-oxaloacetate (OAA) shuttle[27,46,47]. We propose here that stromal ATP is also rapidly consumed in pollen plastids, as ATP is not generated through photosynthesis and the stromal ATP level is lower than the cytosolic ATP level (Fig. 2a–c), even though *NTT2* is highly expressed in growing pollen (Fig. 3a, b). By contrast, our results show that pollen plastids contain higher levels of NADPH and a higher NADH/NAD+ ratio than does the cytosol (Supplementary Fig. 3). Hence, stromal NADPH and NADH are likely to be in surplus compared to ATP in the stoichiometry needed for FA biosynthesis. This hypothesis is supported by

**Fig. 3 | Source of ATP in pollen plastids. a** Histochemical GUS staining of pollen tubes under the control of *AtNTT1* or *AtNTT2* promoters. High GUS activity was detected in pollen tubes and ungerminated pollen grains in three independent lines of *AtNTT2*, but not in *AtNTT1* lines. A WT pollen tube was used as a control. Scale bar = 20 μm. **b** Real-time PCR analysis of the expression of *AtNTT1* and *AtNTT2* in pollen tubes after 4 h germination. EF1α was used as an internal control (*n* = 3 biologically independent samples in each column; error bars ± SEM). Asterisks indicate statistically significant differences (\*\*\**P* < 0.001, two-sided unpaired *t*-test). *p* value of panel b = 1.065 ×10⁻⁶. **c** Schematic representation of T-DNA insertion in *ntt* genes. UTR, untranslated region. The gray number boxes with numbers represent for gene exons. Scale bar = 200 bp. **d**–**g** The knockout of *AtNTT1* and *AtNTT2* in Arabidopsis did not affect pollen tube growth, including length and width (*n* = 100 biologically independent pollen tubes), and pollen germination rate (*n* = 3 biologically independent pollen in each replicate, at least 300 pollen were counted for each value). Error bars ± S.D. of mean value of each replicate. Asterisks indicate statistically significant differences (\*\*\**P* < 0.001, one-way ANOVA with Dunnett's multiple comparison test). *p* values of panel e = 0.591 (WT vs. *ntt1-1*), 0.997 (WT vs. *ntt1-2*), and 0.559 (WT vs. *ntt2*). *p* values of panel f = 0.052 (WT vs. *ntt1-1*), 0.630 (WT vs. *ntt1-2*), and 0.887 (WT vs. *ntt2*). *p* values of panel g = 0.151 (WT vs. *ntt1-1*), 0.340 (WT vs. *ntt1-2*), and 0.501 (WT vs. *ntt2*). Scale bar, 100 μm. **h, i** After introducing the At1.03 to *ntt* knockout lines, the subcellular ATP levels were quantified via FRET ratio in *ntt1-1* and *ntt2* mutants, both in pollen plastids and cytosol. Compared to the WT, no significant differences in ATP level were detected in the growing pollen tubes of mutants (*n* = 5 biologically independent pollen tubes in each column; one-way ANOVA with Dunnett's multiple comparison test, error bars ± SEM). *p* values of panel i = 0.995 (Plastid: WT vs. *ntt1-1*), 0.998 (Plastid: WT vs. *ntt2*), 0.849 (Cytosol: WT vs. *ntt2*), and 0.885 (Cytosol: WT vs. *ntt2*). Scale bar, 20 μm.

the observation that NADP-ME4 can supply substantial amounts of stromal NADPH in pollen (Fig. 4). As the flux of plastid glycolysis is high, much NADH can be generated by plastid-localized GAPCp and PDH (Fig. 8). Both Arabidopsis *GAPCp* genes are highly expressed in pollen[23]. To sustain plastid glycolysis, which supplies the carbon skeletons for FA, surplus stromal NADH must be recycled to NAD⁺ via FA biosynthesis and the reduction of OAA to malate by pNAD-MDH (Fig. 7e–h). In knockout mutants of *pNAD-MDH*, NADH-dependent glutamate synthase (NADH-GOGAT) can also recycle NAD⁺ from NADH when its substrates were added exogenously[36], further supporting that recycling of stromal NAD⁺ from NADH is important for pollen tube growth.

OAA is imported from the cytosol via OAA/malate transporter 1 (OMT1), also named dicarboxylate transporter 1 (DiT1), on the plastid envelope[48]. OAA is synthesized in the cytosol by PEP carboxylase (PEPc)[49]. Arabidopsis pollen grains express a bacterial-type PEPc gene to high levels, and the encoding protein forms the heterooctameric Class-2 PEPc complex. Unlike the heterotetrameric Class 1 PEPc complex, which comprises only plant-type PEPc, the Class 2 PEPc complex is less sensitive to inhibition by the allosteric inhibitor malate[49]. Thermodynamically, the large positive value of ΔG′° of MDH favors the production of malate once OAA is synthesized by PEPc[50], which consumes cytosolic NADH and regenerates NAD⁺. The preferential expression of genes encoding Class-2 PEPc complex components in pollen suggests a high demand for OAA biosynthesis that should not be suppressed by the accumulation of malate in the cytosol. OMT1 can transfer OAA from the cytosol to plastids, but this transporter exchanges OAA for stromal malate. Hence, while OMT1 imports OAA and exports surplus stromal NADH in the form of malate, it cannot supply extra malate to plastid NADP-ME4. The extra malate must be supplied by another transporter, dicarboxylate transporter 2 (DiT2), which exchanges stromal glutamate for cytosolic malate[51,52]. In root plastids, glutamate synthase consumes NADH to synthesize glutamate in the stromal glutamine synthetase/glutamate synthase cycle, which can also partially dissipate surplus NADH in pollen stroma. The imported malate can then be used by NADP-ME4 to generate NADPH and pyruvate in plastids. The flux of transferring NADH into NADPH via the pNAD-MDH/pNADP-ME4 pathway (Fig. 8) is substantial, as stromal NADPH levels are significantly lower in the *nadp-me4* mutant compared to in the WT (Fig. 4e, f), in turn lowering the demand on the OPPP to supply NADPH (Fig. 4). OMT1 is unlikely to export surplus stromal NADPH to the cytosol, as plastid NADP-MDH is inactive in the dark[53,54].

Pollen tube growth consumes a large amount of ATP, and this energy conversion process must be rapid and efficient. Pollen contain many mitochondria[32], which are thought to produce most of the ATP required during tube growth. In tobacco, the respiration rate of pollen is about 10 times higher than that of leaves and can be inhibited by both antimycin A and salicylhydroxamic acid, providing support for the idea that the mitochondrial electron transport chain is the major

site of oxygen consumption in pollen[55]. In lily (*Lilium formosanum*) pollen, treatment with 40 μM oligomycin suppresses oxygen consumption to the detection limit but only transiently inhibits pollen tube growth for a few minutes before growth resumes to 40% of its original rate, without respiration[56]. The observed growth in the presence of oligomycin relies on energy derived from fermentation, as shown by the accumulation of ethanol in the medium[56]. Fermentation is also very active in tobacco and petunia (*Petunia hybrida*) pollen but not in their vegetative tissues[57,58]. Fermentation does not generate ATP directly but recycles NAD⁺ from NADH in the cytosol through the two enzymes PDC and ADH. PDC converts pyruvate into acetaldehyde, which is then converted into ethanol by ADH[58]. Both sets of encoding genes are highly expressed in mature tobacco pollen as well as in germinating pollen under both aerobic and anaerobic conditions, but not in vegetative tissues under normal conditions[55,57]. The recycling of NAD⁺ via ADH is important for sustaining glycolysis, another source of cytosolic ATP. All the above data indicate that both mitochondria and glycolysis supply energy for pollen tube growth, and fermentation is a key process to support pollen tube growth in lily, tobacco, and petunia.

In Arabidopsis, a functional glycolytic pathway is important for pollen tube growth, which is greatly hampered in T-DNA insertion mutants in cytosolic enolase[59] and phosphoglycerate kinase[60]. By contrast, fermentation is not essential for Arabidopsis pollen tube growth. Microarray analyses[61,62] determined that *ADH1* is expressed to very low levels during both in vitro and in vivo germination, and our data confirm that pollen tubes of the *adh1* mutant grow normally (Fig. 6e–h). The differential dependence on fermentation may be correlated to pollen species, as Arabidopsis pollen grains are trinucleate, whereas those of tobacco, lily, and petunia are binucleate[63]. Alternatively, the observed difference in fermentation needs may reflect varying oxygen availability, which would be affected by the size of the pollen tube and the length of the style. Oxygen tension drops from the stigma to the ovary in Amaryllis (*Hippeastrum hybridum*), with a more pronounced effect in styles with high-speed tube growth[64].

Acetaldehyde can be converted into ethanol by ADH or into acetate by ALDHs, which can be further converted into acetyl-CoA by plastid ACS[33]. This pathway, the PDH bypass[12], is not active in vegetative tissues under non-stress conditions. Studies with inhibitors on tobacco pollen tube growth[33] and on the petunia *pdc2* mutant[58] showed that the PDH bypass is important for pollen tube growth in these two species. Hence, we studied the PDH bypass in Arabidopsis pollen in greater detail. We determined that neither gene encoding plastid ALDH enzymes (ALDH3I1[20] and ALDH10A8[21]) is expressed in pollen (Fig. 6i, j). We also showed that the gene encoding cytosolic ALDH7B4 is highly expressed in pollen, but ALDH7B4 appears to be inactive in converting acetaldehyde into acetate[65]. Instead, ALDH7B4 converts ω-aminoaldehydes, particularly α-aminoadipate-semialdehyde, and is involved in the production of cytosolic osmoprotectants[65]. To date, only ALDH family 2 enzymes have been

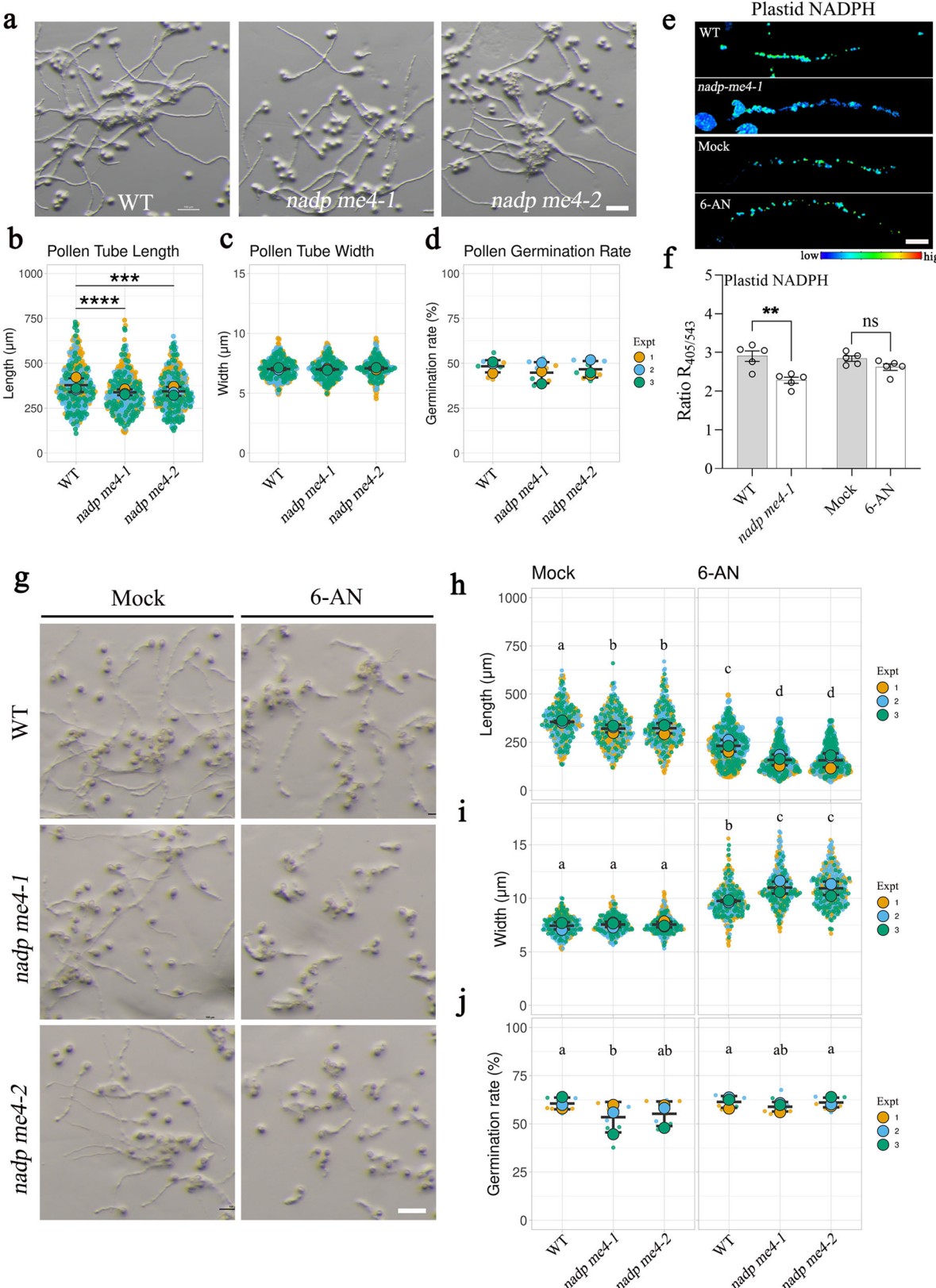

demonstrated to use acetaldehyde as a substrate[33]. The Arabidopsis genome encodes three ALDH2 members. ALDH2C4 is a cytosolic enzyme, while both ALDH2B4 and ALDH2B7 localize to mitochondria. RT-qPCR data indicated that *ALDH2C4* and *ALDH2B4* are not expressed in Arabidopsis pollen, while *ALDH2B7* is highly expressed (Fig. 6i, j). This observation is different from their expression levels in

Arabidopsis seedlings and inflorescences, as both *ALDH2C4* and *ALDH2B4*, but not *ALDH2B7*, are expressed in various seedling tissues[66]. *ALDH2B7* is only expressed in flower buds at moderate levels[66]. No visible growth phenotype was observed in the triple *aldh2c4 aldh2b4 aldh2b7* mutant, indicating that the PDH bypass is not essential for normal Arabidopsis growth[66].

**Fig. 4 | Source of NADPH in pollen plastids. a** In vitro pollen growth assays in WT and homozygous *nadp-me4* T-DNA insertion mutants. **b–d** Pollen tube length, pollen tube width, and pollen germination rate of the WT and the *nadp-me4* mutants. *n* = 100 in pollen length and width comparison of each replicate, *n* = 3 in pollen germination rate measuring (each value was counted from more than 500 pollen). Error bars ± S.D. of mean value of each replicate. Asterisks indicate statistically significant differences (***$P < 0.001$, ordinary one-way ANOVA with Dunnett's multiple comparison test). *p* values of panel b = 5.025 ×10⁻⁵ (WT vs. *nadp me4-1*) and 5.624 ×10⁻⁴ (WT vs. *nadp me4-2*). *p* values of panel c = 0.818 (WT vs. *nadp me4-1*) and 0.621 (WT vs. *nadp me4-2*). *p* values of panel d = 0.251 (WT vs. *nadp me4-1*) and 0.712 (WT vs. *nadp me4-2*). Scale bar, 100 μm. **e** TKTP-mCherry-iNAP4 biosensor in pollen plastids of the WT and *nadp-me4-1* mutant, and the WT pollen treated with 5 mM 6-AN for 30 min are shown. Scale bar, 20 μm. **f** Comparison of NADPH levels in WT and *nadp-me4-1* pollen plastids. *n* = 5, error bars ± SEM. Asterisks indicate statistically significant differences (***$P < 0.001$, two-sided unpaired *t*-test). *p* values of panel f = 0.004 (WT vs. *nadp me4-1*) and 0.077 (mock vs. 6-AN). **g** WT and *nadp-me4* pollen were treated with 5 mM OPPP inhibitor 6-AN. **h–j** The effects of 6-AN treatment on pollen tube length, pollen tube width, and pollen germination rate on the WT and *nadp-me4* mutants. *n* = 100 biologically independent pollen tubes for pollen length and width measurement in each replicate, and more than 500 pollen were counted for pollen germination rate (error bars ± SEM). Asterisks indicate statistically significant differences (***$P < 0.001$, one-way ANOVA with Tukey's multiple comparison test). Exact *p*-values for panel h to j are provided in the source data file. Scale bar, 100 μm.

Arabidopsis has two enzymes that can convert acetate to acetyl-CoA: ACS in plastids and ACETATE NON-UTILIZING 1 (ACN1) in peroxisomes. Both enzymes can prevent the toxic accumulation of acetate, as shown by the normal acetate levels and growth phenotypes seen in each mutant, in contrast to the double mutant lacking the capacity to generate a plastid acetyl-CoA pool in Arabidopsis seedlings[11]. We also observed here that the pollen of both mutants grow normally (Fig. 6e–h). In summary, the PDH bypass, which involves the sequential actions of PDC, ALDH, and ACS, does not appear to play an important role in providing NADH or acetyl-CoA to Arabidopsis pollen plastids. Therefore, we conclude that the plastid PDH pathway is the key pathway to supply acetyl-CoA for FA biosynthesis during pollen tube growth. The stronger inhibitory effect on pollen tube growth of disulfiram compared with AEP in both Arabidopsis (Fig. 6b) and tobacco[33] may be due to its potential inhibition of other ALDHs. Treatment with AEP or disulfiram resulted in a comparable drop in cytosolic and plastid NADH/NAD⁺ ratios, without affecting the NADPH levels in these two compartments, indicating that the NADH/NAD⁺ ratios in plastids, the cytosol, and mitochondria are interconnected[27,67] and that NADH/NAD⁺ ratios and NADPH levels are differentially regulated (Fig. 7a–d). It is well-known that malate valves and citrate circulation play important roles in subcellular shuttle of NADH but not of NADPH[68,69].

As pollen contain many mitochondria[32], and mitochondrial PDH (mPDH) accumulates to high levels in maize and tobacco pollen[70], it is reasonable to assume that mPDH consumes much of the pyruvate generated from glycolysis, which will compete for pyruvate consumption with PDC or any potential plastid import via BASS2[13]. As pyruvate concentrations in pollen plastids and the cytosol are not known, the direction of pyruvate transport is unclear. Nonetheless, our data indicated that the *bass2* mutant pollen grow normally, suggesting that pyruvate transport across the plastid membrane, if occurring, is not essential for pollen tube growth.

Based on the findings of this study, we propose a model of the energy sources supporting pollen tube growth and FA biosynthesis in pollen plastids (Fig. 8).

## Methods

### Development of the second-generation NADH/NAD⁺ biosensor (mCherry-SoNar)

In the first-generation NADH/NAD⁺ biosensor, SoNar, two subunits of the *Thermus aquaticus* Rex family repressor[71] were linked with a circularly permuted yellow fluorescent protein (cpYFP)[72]. The dual excitation wavelengths for SoNar are 405 nm and 488 nm with an emission wavelength of 520 nm. In the second-generation biosensor, mCherry-SoNar, a pH-insensitive fluorophore, mCherry was fused to the N-terminus of the SoNar biosensor with a pentapeptide linker (GGSGG)₄ (Fig. 1a). The new biosensor avoids the pH interference of 488 nm by setting the dual excitation spectra at 405 nm to 420 nm and between 543 nm to 575 nm (depending on the equipment filter set setup), with emissions at 520 nm and 610 nm, respectively.

### Development of the second-generation NADPH sensors (mCherry-iNAPs)

The NADPH biosensors, iNAP1 and iNAP4, are the mutated versions of SoNar in which the positively charged NADPH binding protein residues tightly enriched in the binding pockets of 2'-phosphate and the loop rigidity around the 2' hydroxyl group of the ligand has been reduced. As a result, this favors interactions with NADPH during molecular binding[29]. Similar to the mCherry-SoNar biosensor, the second-generation sensors, mCherry-iNAP1 and mCherry-iNAP4 were created by linking the C-terminus of mCherry and the N-termini of iNAP1 and iNAP4 with the four pentapeptide linkers (GGSGG)₄ (Fig. 1a). The new sensors avoid the pH interference of 488 nm by setting the dual excitation spectra at 405 nm to 420 nm and between 543 nm to 575 nm (depending on the equipment filter set setup), with emissions at 520 nm and 610 nm, respectively.

### Plasmid construction and generation of transgenic plant lines

To construct the plant transformation vectors carrying the second-generation biosensors, the iNAPs and SoNar cDNAs were amplified from the first-generation pENTR- iNAP1, pENTR-iNAP4, and pENTR-SoNar vectors[27] with a 34 bp overhang forward primer carrying a BamH1 restriction site, whereas the mCherry gene was amplified from a pENTR-mCherry-TOC33 vector with a 24 bp overhang reverse primer carrying a BamHI restriction site. This 20-aa peptide linker (GGSGG)₄ was created via the overhang of the forward and reverse primer from SoNar/iNAPs and mCherry, respectively. The mCherry and SoNar/iNAPs cDNAs were ligated and cloned into the EcoRI/HindIII sites of the pRSETb vector. To construct vectors that express the sensors in pollen tubes, the LAT52 promoter was amplified from the pGTkan3-p LAT52 vector, a kind gift from Prof. Li-Qing Chen of the University of Illinois Urbana-Champaign and cloned into the modified Gateway pENTR/D-TOPO vector (Invitrogen, USA) using ClonExpress II One Step Cloning Kit (Vazyme, China). The sequences of the *Nicotiana tabacum* chloroplast transketolase transit peptide (TKTP) (flanked with NdeI/PstI restriction sites)[26], the FRET sensor At1.03[26] (flanked with SpeI/XbaI sites), mCherry-iNAP1, mCherry-iNAP4, mCherry-SoNar (flanked with SpeI/XbaI sites, from the pRSETb fusion constructs) were amplified, purified and individually cloned into a Gateway pENTR/D-TOPO vector carrying the LAT52 promoter. Primer sequences are listed in Supplementary Table 2. All the constructs were confirmed by nucleotide sequencing by BGI, China. All cDNA constructs in pENTR vectors were then transferred to the plant transformation Gateway destination vectors, pEarleyGate302 vector. The pEarleyGate302 sensor vectors were transformed into *Arabidopsis thaliana* (ecotype Columbia, Col-0) using the *Agrobacterium tumefaciens* strain GV3101-mediated floral dip method. Positive transformants were screened using a fluorescence stereomicroscope (Olympus, SZX16).

To generate marker plants for FtsZ1 (At5g55280), the coding sequence of *FtsZ1* was amplified from Arabidopsis cDNA by PCR. The amplicon was cloned into the plant transient expression vector pBI221-mRFP (cleaved by KpnI) or pBI221-mCerulean (cleaved by XbaI) by one-

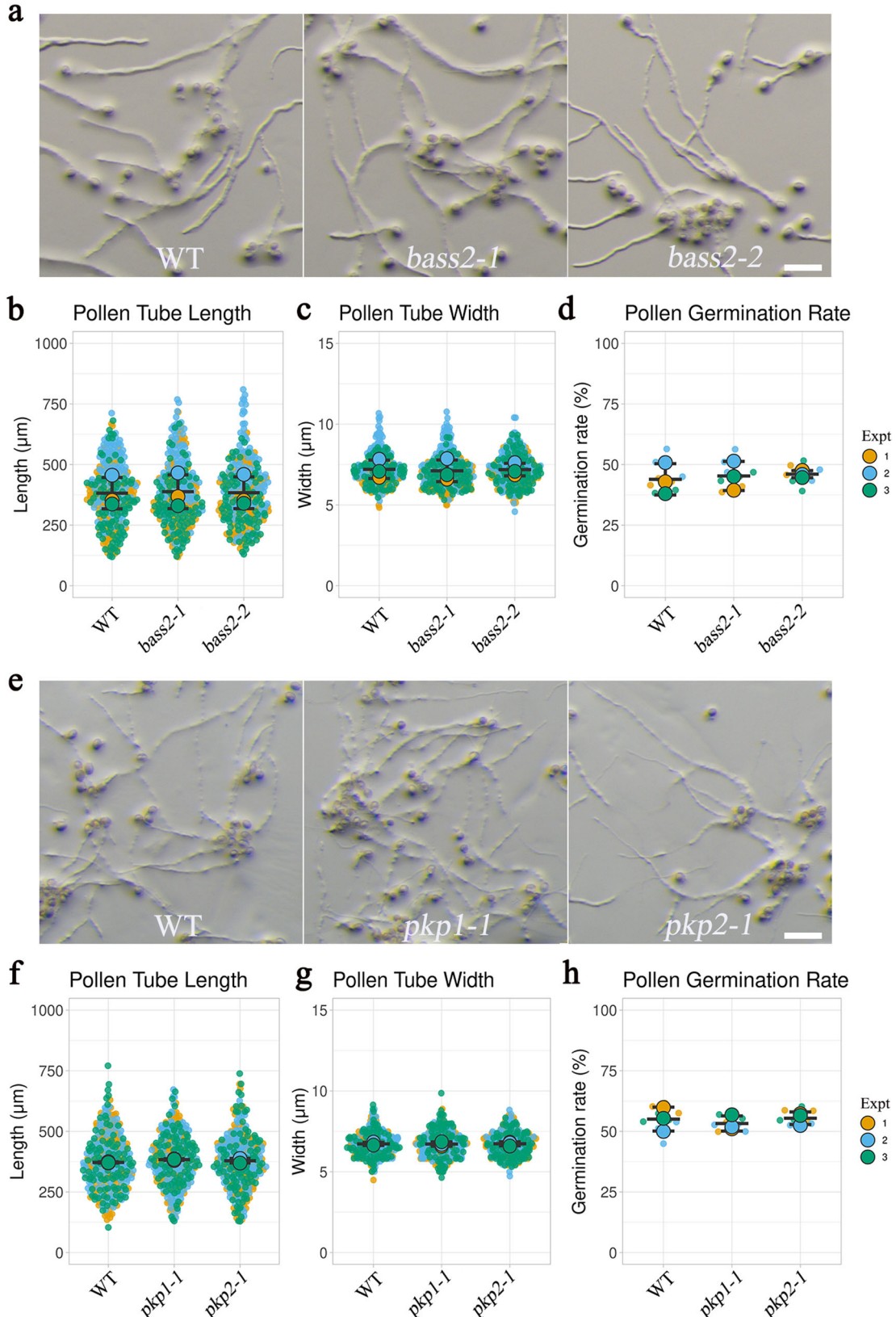

**Fig. 5 | Source of pyruvate in pollen plastids. a** In vitro pollen germination assays for the WT and two homozygous *bass2* T-DNA mutants. **b**–**d** Pollen tube length, pollen tube width, and pollen germination rate of the WT and *bass2* mutants. *p* values of panel b = 0.800 (*bass2-1*) and 0.980 (*bass2-2*). *p* values of panel c = 0.4143 (*bass2-1*) and 0.9621 (*bass2-2*). *p* values of panel d = 0.8125 (*bass2-1*) and 0.6206 (*bass2-2*). **e** Pollen tube growth of WT and *pkp1-1*, *pkp2-1* knockout lines. **f**–**h** Pollen tube length, width and germination rate were compared between WT and *pkp1-1*, *pkp2-1* pollen. *p* values of panel f = 0.352 (*pkp1-1*) and 0.670 (*pkp2-1*). *p* values of panel g = 0.990 (*pkp1-1*) and 0.980 (*pkp2-1*). *p* values of panel h = 0.512 (*pkp1-1*) and 0.972 (*pkp2-1*); *n* = 100 biologically independent pollen tube for each experiment (Expt) on length and width, and *n* = 3 for each replicate on germination rate. Error bars ± S.D. of mean value of each replicate. Significant differences were determined using ordinary one-way ANOVA with Dunnett's multiple comparison test: ***$P$ < 0.001. Scale bar = 100 μm.

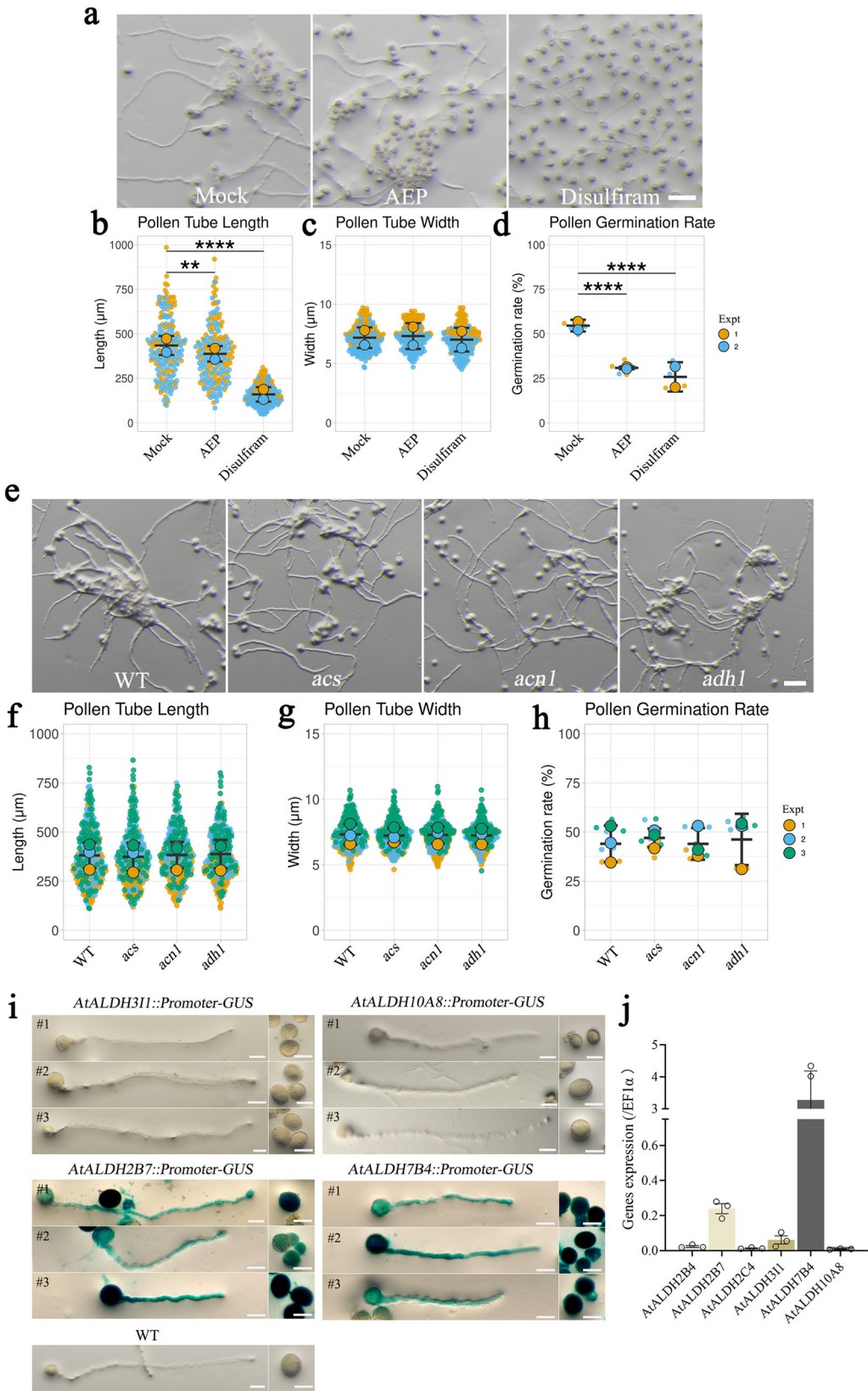

step recombination. The *FtsZ1-mRFP* and *FtsZ1-mCerulean* coding sequences were then cloned into the plant expression vector pBI121::*35S*-GUS (https://www.snapgene.com/resources/plasmid-files/?set=plant_vectors&plasmid=pBI121) with the 35S promoter and *GUS* sequences replaced by the LAT52 promoter (flanked by HindIII/SacI restriction sites). In each step, all PCR-derived constructs were verified

by sequencing. The resulting constructs were introduced into WT plants as mentioned above. Transgenic plants harboring the *FtSZ1-mRFP* or *FtSZ1-mCerulean* transgene were screened with kanamycin antibiotics, and positive plants were screened under an Olympus SZX 16 Stereo fluorescent Microscope. The selected *FtSZ1* plants were crossed to plants carrying the TKTP-fused sensors to get F1

**Fig. 6 | The role of plastid PDH and PDH bypass in pollen tube growth. a** In vitro pollen tube growth of WT pollen subjected to mock medium, or media containing 90 μM AEP or 5 μM disulfiram. **b–d** After incubation for 4 h, pollen tube length, pollen tube width, and pollen germination rate were compared. **e** In vitro pollen tube growth assay of pollen of WT, *acs* (SALK_015522C), *acn1* (SALK_009373C), and *adh1* (SALK_208657C) T-DNA insertion mutants. **f–h** Pollen tube length, pollen tube width, and pollen germination rate of the WT, *acs*, *acn1*, and *adh1* were compared. Error bars ± S.D. of mean value of each replicate. $n = 100$ biologically independent pollen tubes for each experiment (Expt) on length and width, and $n = 3$ for each replicate on germination rate. *p* values of b = $1.747 \times 10^{-3}$ (mock vs. AEP) and $3.469 \times 10^{-8}$ (WT vs. disulfiram). *p* values of c = 0.339 (mock vs. AEP) and 0.2237 (WT vs.

disulfiram). *p* values of d = $6.022 \times 10^{-7}$ (mock vs. AEP) and $7.789 \times 10^{-8}$ (WT vs. disulfiram). *p* values of f = 0.794 (WT vs. *acs*) and 0.995 (WT vs. *acn1*), and 0.850 (WT vs. *adh1*). *p* values of g = 0.389 (WT vs. *acs*), 0.943 (WT vs. *acn1*), and 0.616 (WT vs. *adh1*). *p* values of h = 0.804 (WT vs. *acs*), >0.999 (WT vs. *acn1*), and 0.907 (WT vs. *adh1*). Asterisks indicate significant differences: ***$P < 0.001$, by ordinary one-way ANOVA with Dunnett's multiple comparison test in b–d, f–h. Scale bar = 100 μm. **i** Histochemical GUS staining of pollen tubes of *AtALDH3I1pro*: GUS, *AtALDH10A8-pro*:GUS, *AtALDH2B7pro*:GUS, and *AtALDH7B4pro*:GUS plants, with the WT as a control. **j** qRT-PCR analyses of RNA abundance of various *AtALDHs* in WT pollen tubes, using *EF1α* as a loading control.

hybrids. Primer sequences used for cloning are listed in Supplementary Table 2.

Transgenic plants carrying promoter fusions driving the expression of the *GUS* reporter gene used in this study included *NTT1pro:GUS*, *NTT2pro:GUS*, *ALDH3I1pro:GUS*, *ALDH10A8pro:GUS*, *ALDH2B7pro:GUS*, and *ALDH7B4pro:GUS* (Supplementary Table 3). The *NTT* and *ALDH* promoter regions were PCR amplified from Arabidopsis Col-0 genomic DNA. The PCR products for *NTT1* (1.5 kb), *NTT2* (1.5 kb), *ALDH3I1* (0.9 kb), *ALDH10A8* (1.3 kb), *ALDH2B7* (1.9 kb), and *ALDH7B4* (0.8 kb) promoters were then cloned into the upstream region of the *GUS* gene in the binary vector pBI121::*35S-GUS,* from which the 35S promoter was removed by HindIII and XmaI restriction digestion. All amplified promoter sequences were confirmed by sequencing. All primers used in the construction of the *GUS* vectors are listed in Supplementary Table 2.

## mCherry-SoNar/iNAPs recombinant protein expression and purification

*Escherichia coli* BL21 (DE3) pLysS carrying the pRSETb mCherry-SoNar, mCherry-iNAP1, and mCherry-iNAP4 expression plasmids were grown in a 15 mL LB medium containing 100 μg mL$^{-1}$ (w/v) carbenicillin at 37 °C overnight. A 15 mL starter culture was then subcultured into 150 mL of LB medium containing 100 μg mL$^{-1}$ (w/v) carbenicillin at 37 °C until the absorbance at 600 nm reached approximately 0.6. Then 0.1 mM isopropyl β-D-1-thiogalactopyranoside (IPTG) was used to induce the expression of the His-tagged proteins, and the cells were grown for 16 h at 18 °C. Cells were pelleted by centrifugation at 4000 × g for 30 min at 4 °C, and suspended in ice cold washing buffer (20 mM sodium phosphate buffer, pH 7.4, containing 0.5 M sodium chloride, 40 mM imidazole, and cOmplete protease inhibitor cocktail (Roche, Germany)) and lysed by sonication. The lysates were fractionated by centrifugation at 16,000 × g for 5 min at 4 °C. The supernatants were loaded into a 1 mL HisTrap™ FF column (GE Healthcare, USA) with a 5 mL syringe attached to a 0.45 μm filter. After washing with 2 column volumes of washing buffer, the proteins were eluted from the resin using elution buffer (20 mM sodium phosphate buffer, pH 7.4, containing 0.5 M sodium chloride, and 400 mM imidazole). The recombinant proteins were dialyzed overnight in dialysis buffer (20 mM Tris-HCl at pH 7.5 and 150 mM sodium chloride) and stored at −80 °C prior to the assay. The elutes were separated by 10% (w/v) SDS-PAGE and stained with Coomassie blue dye. NativeMark™ unstained protein standard (Invitrogen, USA) was used as a molecular weight marker. The molecular weights of mCherry-SoNar, mCherry-iNAP1, and mCherry-iNAP4 proteins were detected at 69.7 kDa.

## Plant materials and growth conditions
Arabidopsis (*Arabidopsis thaliana*) seeds were surface-sterilized with 20% (v/v) Clorox for 15 min and washed with distilled water three times before germinating on full-strength Murashige and Skoog medium[73] supplemented with 2% (w/v) sucrose, 0.1% (w/v) 2-(*N*-morpholino) ethanesulfonic acid, and 0.25% (w/v) Phytagel with a pH adjusted to

5.7–5.8 with KOH. After 2 days in darkness at 4 °C, plates were moved to the tissue culture room for seed germination. Ten-day-old seedlings were then transferred to the soil. Plants were grown in the growth chamber under a photoperiod of 12 h light/12 h dark with a temperature cycle of 22 °C (day) and 18 °C (night) until they reached the flowering stage for pollen analyses. All WT plants used in this study were in the Columbia (Col-0) accession.

## Multiwell plate reader-based fluorimetry
Fluorescent intensities of purified mCherry-SoNar and mCherry-iNAPs recombinant protein and true leaves were collected using the Cytation 1 Cell Imaging Multi-mode Reader (BioTek, USA). mCherry-SoNar and mCherry-iNAPs were excited with dual wavelengths of 400 ± 10 nm and 580 ± 25 nm, and the emission wavelength was recorded at 520 ± 25 nm and 620 ± 10 nm using a black 96-well microplate (Corning Costar, USA). The internal temperature was kept at 25 °C.

## In vitro characterization of mCherry-SoNar/iNAPs
For in vitro calibration of the protein sensors, protein concentrations of purified mCherry-SoNar and mCherry-iNAPs were quantified using a Bradford protein assay (Bio-Rad, USA). To determine the $K_d$ values of the recombinant protein sensors, the purified mCherry-SoNar and mCherry-iNAPs proteins were titrated using gradient increment of NADPH, NADH, and NAD$^+$ nucleotide concentrations at different pH values at 7.0, 7.5, 8.0, and 8.5. NADPH concentrations were set at 0 μM, 0.001 μM, 0.01 μM, 0.05 μM, 0.1 μM, 0.5 μM, 1 μM, 2 μM, 4 μM, 5 μM, 8 μM, 10 μM, 20 μM, 25 μM, 50 μM, 62.5 μM, 100 μM, 125 μM, 150 μM, 200 μM, 250 μM, 400 μM, and 500 μM, whereas the ratios of 100 μM NADH and NAD$^+$ were set as 0, 0.0008, 0.001, 0.002, 0.004, 0.007, 0.01, 0.014, 0.02, 0.04, 0.06, 0.11, 0.25, 0.50, 0.75, 1, 1.25, 1.4, 1.7, 1.9, 2.25, 2.5, 3.25, and 3.75. The purified mCherry-SoNar and mCherry-iNAPs proteins were diluted in pseudocytosol medium containing 100 mM potassium gluconate, 30 mM NaCl, 25 mM MES, 25 mM HEPES, 40% sucrose, and 1 mg mL$^{-1}$ (w/v) BSA (pH 7.5) to the concentration of 0.5 μM. Each assay was performed with 50 μL pyridine nucleotides and 50 μL protein arrayed in a black 96-well microplate. Fluorescence characteristics of purified mCherry-SoNar and mCherry-iNAPs were detected by Cytation 1 multi-mode reader as aforementioned.

To evaluate the effects of various inhibitors, oxidants, reductant, and antioxidants on sensor ratios, 0.5 μM purified mCherry-SoNar or mCherry-iNAPs sensors in 50 μL pseudocytosol medium pH 7.5 and 50 μL of inhibitors (50 μM rotenone, 100 μM thenoyltrifluoroacetone (TTFA), 10 μM antimycin A, 10 μM oligomycin, 500 μM potassium cyanide (KCN), 90 μM 1-aminoethylphosphonic acid (AEP), 30 μM disulfiram, 40 μM CGP3466B maleate, 100 μM iodoacetate, 2 mM salicylhydroxamic acid (SHAM)), oxidants (1 mM hydrogen peroxide (H$_2$O$_2$) and 30 μM menadione), reductant (10 mM dithiothreitol (DTT)), and antioxidants (10 mM L-ascorbic acid or 10 mM reduced glutathione) were added into a black 96-well microplate. The fluorescence intensities of purified mCherry-SoNar and mCherry-iNAPs were collected by the Cytation 1 multi-mode reader.

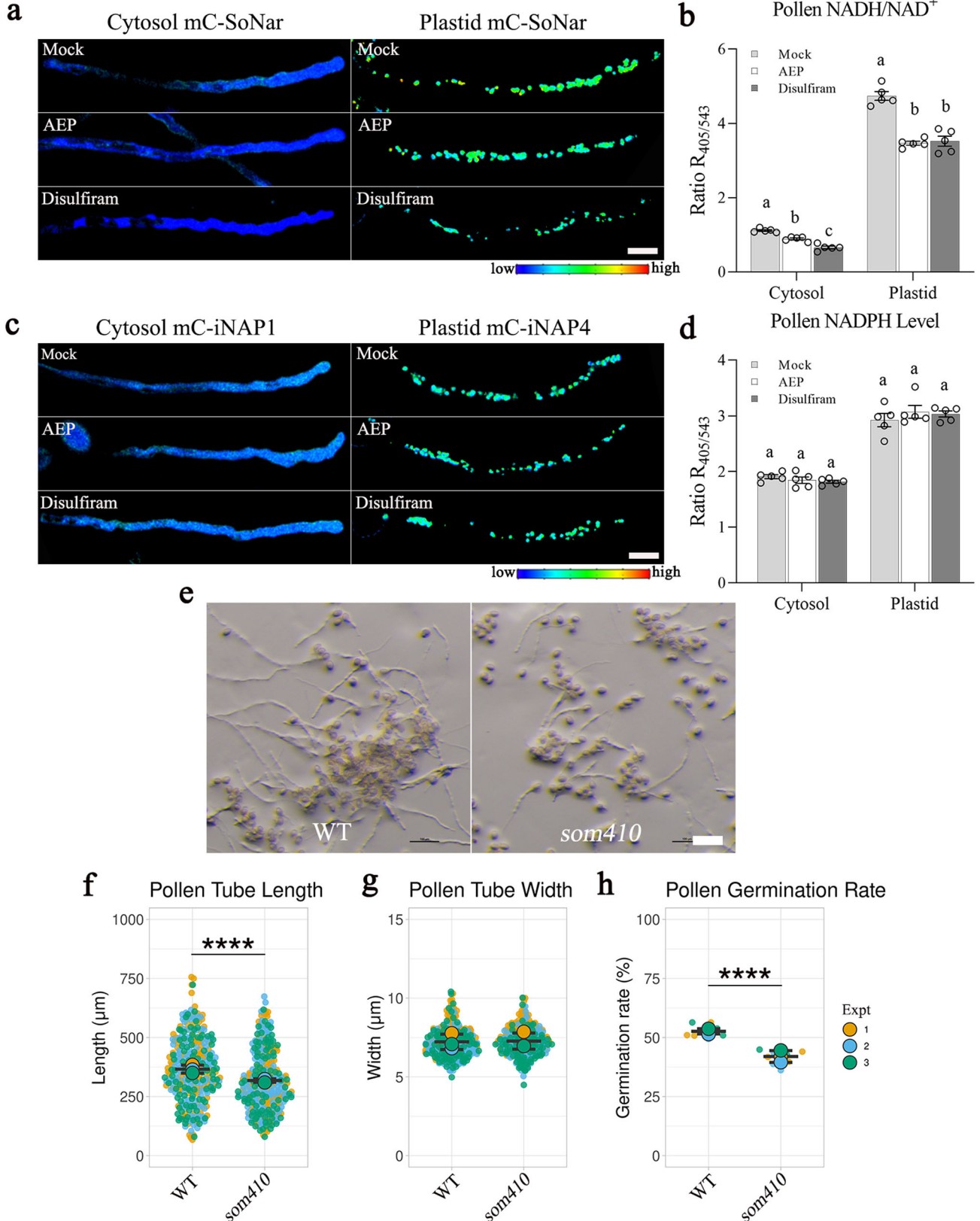

### In vitro pollen germination and quantification

In vitro pollen germination was performed as described[74]. Freshly open Arabidopsis flowers at stage 13 were harvested, and mature pollen grains were dusted on the surface of solid pollen germination medium containing 1 mM $Ca(NO)_3$, 1 mM $Ca(Cl)_2$, 1 mM $MgSO_4$, 0.01% (w/v) boric acid, 18% (w/v) sucrose, and 0.8% (w/v) agarose, with the pH adjusted to 7.5 with KOH, which was put on glass slides. The slides covered with germination gel and pollen grains were placed in square petri dishes with wet filter paper to maintain the humidity, sealed, and incubated in a water bath at 28 °C for 4 h. To observe pollen tubes, gel pieces with germinating pollen were visualized under an Olympus SZX 16 Stereo fluorescent Microscope equipped with an achromatic 1×

**Fig. 7 | Source of NADH in pollen plastids. a-d** After pollen germinated on solid pollen germination medium, pollen tubes of cytosol mCherry-SoNar, plastid mCherry-SoNar, cytosol mCherry-iNAP1 and plastid mCherry-iNAP4 were treated with 300 μM AEP or 50 μM disulfiram for 30 min. Emissions of 535 ± 40 nm excited at 405 nm, and 630 ± 60 nm excited at 543 nm were recorded. $n = 5$ biologically independent pollen in each column in b, d, error bars ± SEM. Asterisks indicate statistically significant differences (***$P < 0.001$, one-way ANOVA with Tukey's multiple comparison test). $p$ values of b = 4.941 ×10$^{-3}$ (cytosol: mock vs. AEP), 4.396 ×10$^{-7}$ (cytosol: mock vs. disulfiram), 3.906 ×10$^{-4}$ (cytosol: AEP vs. disulfiram), 6.645 ×10$^{-6}$ (plastid: Mock vs. AEP), 1.027 ×10$^{-5}$ (plastid: mock vs. disulfiram), and 0.936 (plastid: AEP vs. disulfiram). $p$ values of d = 0.563 (cytosol: mock vs. AEP), 0.322

(cytosol: mock vs. disulfiram), 0.889 (cytosol: AEP vs. disulfiram), 0.562 (plastid: mock vs. AEP), 0.725 (plastid: mock vs. disulfiram), and 0.960 (plastid: AEP vs. disulfiram). Scale bar = 20 μm. **e-h** In vitro pollen growth assay of the WT and *som410* mutant. Pollen tube length, pollen tube width, and pollen germination rate were measured after 4 h pollen incubation. $p$ value of f = 1.374 ×10$^{-5}$. $p$ value of g = 0.532. $p$ value of h = 6.328 ×10$^{-7}$. $n = 100$ biologically independent pollen tubes for pollen length and width measurement in each experiment (Expt), and $n = 3$ (more than 500 pollen were counted) for pollen germination rate, error bars ± S.D. of mean value of each replicate. Asterisks indicate statistically significant differences (***$P < 0.001$, two-sided unpaired $t$-test). Scale bar = 100 μm. mC mCherry.

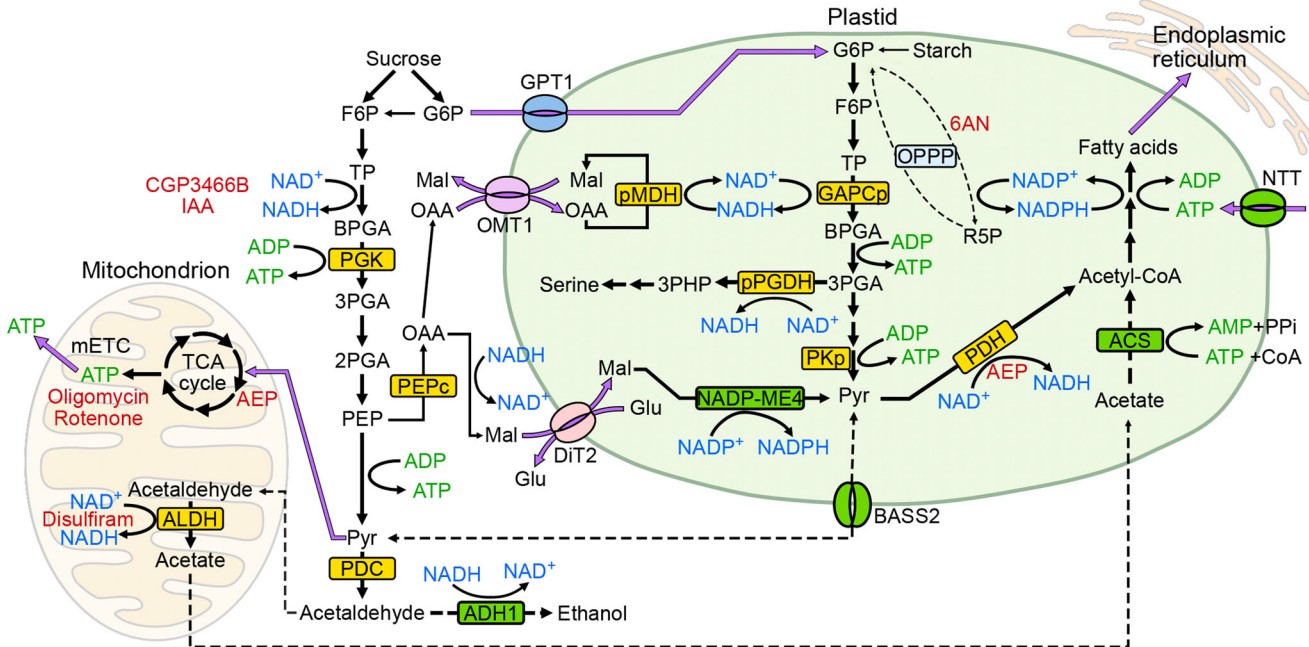

**Fig. 8 | Model of the bioenergetics of Arabidopsis pollen tube growth.** Exogenous sucrose is the major source of energy, reducing equivalents and carbon skeleton for pollen tube growth. In the cytosol, F6P derived from sucrose is metabolized via glycolysis to generate ATP, NADH, PEP, and pyruvate. In addition to pyruvate, some of the PEP is converted into OAA by PEPc, which can be further reduced to malate. While pyruvate and malate can be used as substrates for mitochondria to generate ATP, cytosolic malate can also be imported into plastids via DiT2, which in turn is converted into pyruvate by NADP-ME4 to supply NADPH for FA synthesis. Cytosolic G6P derived from sucrose is imported into plastids via GPT1, which is broken down into pyruvate and generates ATP and NADH via plastid glycolysis. Some imported G6P can also be used to generate NADPH via the OPPP pathway, although the NADP-ME4 pathway remains the major source of plastid NADPH. Unlike tobacco, PDH rather than the PDH bypass is the major source of acetyl-CoA and NADH for FA synthesis in Arabidopsis plastids. In addition to plastid glycolysis, cytosolic ATP is also imported into the stroma via the NTT2 transporter. Unlike tobacco and lily, fermentation is not essential for Arabidopsis pollen tube growth. Inhibitors and mutants employed in this study are highlighted in red and in

green boxes. Solid black and purple lines represent major metabolic pathways and metabolite transports, respectively. Broken lines represent metabolic pathways that do not play important roles in Arabidopsis pollen tube growth. ACS, acetyl-CoA synthase; ADH1, alcohol dehydrogenase 1; ALDH, aldehyde dehydrogenase; BASS, bile acid:sodium symporter family protein 2; BPGA, bisphosphoglyceric acid; DiT2, dicarboxylate transporter 2; F6P, fructose 6-phosphate; Glu, glutamate; G6P, glucose-6-phosphate; GPT1, glucose-6-phosphate/Pi transporter 1; Mal, malate; mETC, mitochondrial electron transport chain; NADP-ME4, NADP-dependent malic enzyme 4; NTT, nucleotide transporter; OAA, oxaloacetate; OMT1, oxaloacetate/malate transporter 1; OPPP, oxidative pentose-phosphate pathway; PDC, pyruvate decarboxylase; PDH, pyruvate dehydrogenase; PEP, phosphoenolpyruvate; PEPc, phospho*enol*pyruvate carboxylase; 2PGA, 2-phosphoglycerate; 3PGA, 3-phosphoglycerate; PGK, phosphoglycerate kinase; 3PHP, 3-phosphohydroxypyruvate; PKp, plastid pyruvate kinase; pMDH, plastid NAD-dependent malate dehydrogenase; pPGDH, plastid 3-phosphoglycerate dehydrogenase; Pyr, pyruvate; R5P, ribose-5-phosphate; TCA, tricarboxylic acid; TP, triose phosphate.

Stereo Objective, a 10× 0.45 NA objective, and a 20× 0.60 NA objective. Images were captured with a Nikon Digital F13 color camera system. Pollen germination rate, pollen tube length, and width were quantified with Fiji software (https://imagej.net/Fiji).

**GUS histochemical staining**
Pollen grains were harvested and germinated on the surface of solid pollen germination medium in a 24-well plate for 4 h at 28 °C. Pollen tubes were then fixed in 90% (v/v) acetone for 20 min at 4 °C and washed three times with washing buffer (25 mM NaH$_2$PO$_4$, pH 7.2, 25 mM Na$_2$HPO$_4$, pH 7.2, 0.1% [v/v] Triton X-100, 2 mM K$_4$[Fe(CN)$_6$],

2 mM K$_3$Fe(CN)$_6$, and 5 mM Na$_2$-EDTA), followed by vacuum infiltration for 20 min on ice with 2 mM X-Gluc (Sigma 203783) dissolved in the washing buffer. After overnight incubation at 37 °C in the dark, pollen tubes were washed with 70% (v/v) ethanol for destaining and examined using an Olympus SZX 16 Stereo fluorescent Microscope.

**Mutant verification**
Seeds for the *nadp-me4-1* (SALK-064163) and *nadp-me4-2* (GK371F05) mutants were kindly provided by Prof. Bentsink of Wageningen University[75]. Seeds for *bass2-1* (SALK_101808) and *bass2-2* (SALK_098962) were kind gifts from Prof. Furumoto of Ryukoku

University[13]. Seeds for *som410* were kindly provided by Prof. Li of the University of Chinese Academy of Sciences[7]. Seeds of the *ntt1/2* double knockout line were kindly provided by Prof. Ekkehard Neuhaus of Universität Kaiserslautern[76]. Seeds of the *pkp1-1* (EAL11) and *pkp2-1* (FCM8) were gifts from Prof. Sébastien Baud of Université Paris-Saclay[77]. Seeds for *ntt1-1* (SALK_083518C), *ntt1-2* (SALK_023159C), *ntt2* (SALK_031126C), *adh1* (SALK_208657C), *acn1* (SALK_009373C), and *acs* (SALK_015522C) were obtained from the ABRC (https://abrc.osu.edu/). Homozygous plants for the mutants obtained from the ABRC were identified by genotyping PCR with appropriate primers, designed on the SIGnAL website (http://signal.salk.edu/tdnaprimers.2.html) (Supplementary Table 2). The precise position of each T-DNA insertion was determined by DNA sequencing of the PCR products (Supplementary Table 2).

### Inhibitor treatment

To study the effects of inhibitors on pollen germination and growth, pollen was dusted onto the surface of solid pollen germination medium containing the relevant inhibitor at the indicated concentrations (5 nM oligomycin, 40 μM CGP3466B, 1 μM iodoacetate, 5 mM 6-AN, 90 μM AEP[78], and 5 μM disulfiram[33]), then imaged under the Olympus SZX 16 Stereo fluorescent Microscope. To study the effects of exogenous application of inhibitors on real-time changes of biosensor ratios, inhibitors were dissolved in liquid pollen germination medium at the indicated concentrations (40 μM oligomycin, 50 μM rotenone, 200 μM CGP3466B, 100 μM iodoacetate, 5 mM 6-AN, 300 μM AEP, 40 μM disulfiram) and added to the surface of pollen tube germinating gel pieces. After 5 min vacuum infiltration followed by incubation for another 25 min, the biosensor signals were recorded with a Zeiss LSM 710 confocal microscope.

### RNA isolation and RT-qPCR

Total RNA was extracted from the cultivated pollen tubes using Trizol reagent (Invitrogen 15596026). Removal of contaminated genomic DNA and first-strand cDNA synthesis were conducted with the HiScript® III 1st Strand cDNA Synthesis Kit (Vazyme R312) according to the manufacturer's instructions. First-strand cDNA was obtained from 1 μg total RNA in a 20 μl reaction system, and 0.5 μl (25 ng RNA equivalent) was used as the template for qPCR, which was performed using SYBR green premix reagent (Applied Biosystems 4367659) in the StepOnePlus™ Real-Time PCR System (Applied Biosystems 4376600). The Arabidopsis housekeeping genes *EF1α* (At1g07940), *GAPDH* (At1g13440), and *UBC9* (At4g27960) were used as the internal controls. Their RT-qPCR primer sequences are listed in Supplementary Table 2.

### Confocal microscopy and image analysis

After 4 h incubation at 28 °C, pollen tubes expressing various biosensor constructs were observed using a Zeiss LSM 710 confocal microscope with a 40× oil-immersion objective lens. Filter sets for pollen tubes expressing (TKTP-)mCherry-iNAP1/iNAP4/SoNar were as follows: excitation 405 nm/emission 535 ± 40 nm for iNAP1/iNAP4/SoNar; excitation 543 nm/emission 630 ± 60 nm for mCherry. The ratios represented by $R_{405/543}$ were calculated as the fluorescence intensity (emission at 535 ± 40 nm) upon excitation at 405 nm divided by the fluorescence intensity (emission at 630 ± 60 nm) upon excitation at 543 nm after background subtraction and autofluorescence correction. At1.03 was excited at 458 nm, and emission was detected at 488 ± 18 nm for mseCFP images and 535 ± 10 nm for FRET images. The biosensor ratio of confocal images was calculated on a pixel-by-pixel basis using the redox analysis software RRA (https://markfricker.org/77-2/software/redox-ratio-analysis/redox-ratio-analysis-software-download/), running in a custom MatLab (https://www.mathworks.com/products/matlab.html) analysis suite. The images were analyzed with (x, y) noise filtering, background subtraction, and autofluorescence correction and

exported as pseudocolor HSV ratio images. For mCherry-iNAP1/iNAP4/SoNar, the autofluorescence correction was obtained by excitation at 405 nm, with emission at 450 ± 19 nm.

### Data analysis

Data plotted using GraphPad Prism (version 9) are presented as means ± standard error of the mean (SEM) in Figs. 1b–e, 2b, c, e, k, 3b, i, 4f, 6j, 7b, d, and Supplementary Figures 1-4 and 8, respectively. Data plotted using SuperPlotsOfData app (https://www.molbiolcell.org/doi/10.1091/mbc.E20-09-0583)[79] are presented as means ± standard deviation (S.D) of mean value of each replicate in Figs. 2g–i, 3e–g, 4b–d, h–j, 5b–d, 5f–h, 6b–d, f–h, 7f–h, and Supplementary Fig. 7, respectively. Statistical significance between contrasting groups was determined by analysis of variance (ANOVA) with Tukey's HSD test, Dunnett's multiple comparison test, two-sided unpaired *t*-tests, or two sided t-test at $P < 0.001$, $P < 0.01$, and $P < 0.05$ in GraphPad Prism (version 9) or SPSS (version 28). *P* value of Dunnett's multiple comparison test were adjusted *P* value.

### Accession numbers

Sequence data used in this study can be found in the Arabidopsis Information Resource (https://www.arabidopsis.org) under the following accession numbers: *NTT1* (At1g80300), *NTT2* (At1g15500), *ALDH3I1* (At4g34240), *ALDH10A8* (At1g74920), *ALDH2B7* (At1g23800), *ALDH7B4* (At1g54100), *FtsZ1* (At5g55280).

### Reporting summary

Further information on research design is available in the Nature Portfolio Reporting Summary linked to this article.

## Data availability

The authors declare that the main data supporting the findings of this study are available within the article and its Supplementary Information files. Source data are provided with this paper. Image data will be uploaded to the repository Biostudies. The accession number is S-BIAD593 (https://www.ebi.ac.uk/biostudies/bioimages/studies/S-BIAD593?key=21ad2704-2e89-4922-885b-8d6d61df898e). Additional data are available from the corresponding author upon request. Source data are provided with this paper.

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

## Acknowledgements

This research is funded by the Hong Kong Research Grants Council General Research Fund (17103921) and Area of Excellence Scheme (AoE/M-403/16), the National Natural Science Foundation of China (32070394), HKU Seed Fund for Basic Research (202111159134) and the Innovation and Technology Fund (Funding Support to State Key Laboratory of Agrobiotechnology) of the Hong Kong Special Administrative Region, China. Any opinions, findings, conclusions, or recommendations expressed in this publication do not reflect the views of the Government of the Hong Kong Special Administrative Region or the Innovation and Technology Commission.

## Author contributions

B.L.L. conceptualized the research. B.L.L. and J.L. designed the study. S.L.L. and J.Y.Z. produced the mCherry-iNAP and mCherry-SoNar transgenic lines. J.L. carried out all the pollen experiments. S.L.L. carried out all the purified protein experiments. J.L. and S.L.L. analyzed the data. B.L.L. and J.L. wrote the manuscript. All authors revised and approved the manuscript.

## Competing interests

The authors declare no competing interests.
