## [Peer Review File · Nature Communications]

Bioenergetics of pollen tube growth in *Arabidopsis thaliana* revealed by ratiometric genetically encoded biosensorsREVIEWER COMMENTS

Reviewer #1 (Remarks to the Author):

“Bioenergetics of pollen tube growth in *Arabidopsis thaliana* revealed by ratiometric genetically encoded biosensors” by Jinhong Liu, Shey-Li Lim, Jia Yi Zhong and Boon Leong Lim

Nature Communications Research Article

MS ID: NCOMMS-22-21551A-Z

The authors present a research article about energy generation supporting pollen tube growth in *Arabidopsis thaliana*.

The authors present an interesting manuscript and the content has been well described. In addition, the research described here is of utmost importance to understand energy generation during pollen tube development. Knowledge that is lacking so far.

In general, the authors need to improve their language (e.g. grammar). Furthermore, original research papers need to be cited and various references are lacking. Furthermore, there are several points which need to be paid attention to:

Minor points:

1. Lines 183-184: I guess you refer to Fig. 3 c-g, not Fig. 2 c-g.
2. Lines 317-318: OAA is synthesized in the cytosol by PEP carboxylase (PEPc) 47. What about cytosolic MDHs? These enzymes can also generate OAA in the cytosol.
3. Lines 402-406: Treatment with AEP or disulfiram resulted in a comparable drop in cytosolic and plastid NADH/NAD⁺ ratios, without affecting the NADPH levels in these two compartments, indicating that the NADH/NAD⁺ ratios in plastids, the cytosol, and mitochondria are interconnected 25, 44 and that NADH/NAD⁺ ratios and NADPH levels are differentially regulated (Fig. 7a–d). In this context, malate valves and citrate circulation should be mentioned as possible routes. It is well known that plastids and mitochondria are interconnected via the cytosol. And that metabolite shuttling plays an essential role.
4. Lines 487-491: The PCR products for NTT1 (1.5 kb), NTT2 (1.5 kb), ALDH3I1 (0.9 kb), ALDH10A8 (1.3 kb), ALDH2B7 (1.9 kb), and ALDH7B4 (0.8 kb) promoters were then cloned upstream of the GUS gene in the binary vector pBI121::35S-GUS with the 35S promoter removed by HindIII and XmaI restriction digestion. The promoter length of ALDH3I1 and ALDH7B4 is quite small. Why have the authors chosen such a short promoter region? In such a short promoter, various regulatory elements might be missing.
5. Lines 516-519: *Arabidopsis* (*Arabidopsis thaliana*) seeds were surface-sterilized with 20% (v/v) Clorox for 15 min and washed with distilled water three times before germinating on Murashige and Skoog medium 68 supplemented with 2% (w/v) sucrose, 0.1% (w/v) 2-(N-morpholino) ethanesulfonic acid, and 0.25% (w/v) Phytigel with a pH adjusted to 5.7–5.8 with KOH. ½ MS medium? Or full-strength MS medium?
6. Fig. 3 c-g: Why haven't the authors used already published knockout lines? And what about the analysis of the double mutant? This is available.
7. Discussion: It has been published that pNAD-MDH interacts with the FtsH12/FtsHi complex and Ycf2 which play an essential role in plastid development. This needs to be added to the discussion.

Major points:

8. Lines 221-222: Among the three plastid pyruvate kinase genes in *Arabidopsis*, PKp1 was highly expressed in pollen tubes in a histochemical GUS assay 31. Hence, we conclude that both NADP-ME4 and plastid glycolysis supply pyruvate to pollen plastids. This needs to be shown, not only concluded. Why didn't the authors analyze pollen tube growth of a *PKp1* mutant?
9. Lines 238-249: In the PDH bypass, acetaldehyde is converted to acetate by ALDH 33. To examine

whether this enzymatic step takes place within pollen plastids or in any other compartments, we evaluated the expression profiles of several Arabidopsis ALDHs in pollen tubes by reverse transcription–quantitative PCR (RT-qPCR) or GUS histochemical assays. We established that the genes encoding the only two plastid-localized ALDHs (chloroplast-localized ALDH31 and leucoplast-localized ALDH10A8) were not expressed in pollen tubes (Fig. 6i, j). Among other ALDHs that may participate in the acetaldehyde–acetate reaction, the genes encoding cytosolic ALDH7B4 and mitochondrial ALDH2B7 were highly expressed in growing pollen tubes, as shown by RT-qPCR and GUS staining (Fig. 6i, j). Hence, acetaldehyde is likely to be converted to acetate by ALDH2B7 in mitochondria but not in plastids. The lack of growth defects in the *acs* mutant also indicated that the PDH bypass is not essential for Arabidopsis pollen tube growth, making the plastid PDH pathway the key source of acetyl-CoA for FA biosynthesis.

In the introduction, the authors describe that there are 16 genes encoding ALDHs but only 4 have been analyzed via GUS staining and 6 via qRT-PCR. At least, gene expression of all 16 ALDHs in pollen tubes should be analyzed.

10. Lines 252-253: There are four possible sources of NADH in non-photosynthetic plastids: plastid glycolysis, pPDH, pNAD-MDH 7, and plastid ALDH 23.

That's not true. What about NADH-GOGAT, GapA/B and PGDH? These have not been considered.

11. Lines 259-262: pNAD-MDH is likely to generate malate and NAD⁺ instead of NADH in plastids to support plastid glycolysis and supply malate for NADP-ME4. We determined that this step is important, as a mutant (suppressor of *mod1* 410 [*som410*]) 7 harboring an A90V amino acid substitution in pNAD-MDH was characterized by slightly shorter pollen tubes (Fig. 7e–h).

It has already been shown in the literature that pNAD-MDH is highly expressed in pollen and pollen tubes. Furthermore, 50% of the pollen of heterozygous knockout mutants do not develop a pollen tube. However, by adding NADH-GOGAT substrates pollen tube growth could be rescued indicating an important role of NADH-GOGAT in pollen tube growth which has not been considered in this manuscript.

12. Lines 398-400: Therefore, we conclude that the plastid PDH pathway is the key pathway to supply acetyl-CoA for FA biosynthesis during pollen tube growth.

Again, it's only a conclusion but the authors do not present any data to show that.

13. Lines 400-402: The stronger inhibitory effect on pollen tube growth of disulfiram compared with AEP in both Arabidopsis (Fig. 6b) and tobacco 32 may be due to its potential inhibition of other ALDHs.

Therefore, the authors should at least analyze transcript levels of other ALDHs (see above).

14. Lines 615-616: The Arabidopsis housekeeping gene EF1 α (At5g60390) was used as the internal control.

At least 2 housekeeping genes need to be used. That's standard today. One housekeeping gene is not enough.

Reviewer #2 (Remarks to the Author):

Jinhong Liu and colleagues modify a number of biosensors and apply those to study the energy fluxes during pollen tube growth in Arabidopsis thaliana. My review focuses on the engineering, characterisation and application of the biosensors (and not on energy metabolism or plant biology) as this is my area of expertise.

The biosensor approach is suitable, but the biosensors performance needs a more thorough validation when comparing different localisations (e.g. plastids vs. cytoplasm). This issue is, together with other suggestions for improvement, further explained below.

Point 1:

The claim that 'pH insensitive' second-generation ratiometric biosensors for pyridine nucleotides were developed is an overstatement. Only a rather narrow range of pH 7.0-8.5 was tested. This statement should be downtrend accordingly.

Point 2:

The notion that a pH insensitive sensor for iNap can be made by fusing it to mCherry was already mentioned in the original paper: “Fusion of iNap1 and the red fluorescent protein mCherry allows ratiometric and pH-resistant measurement”. And this is documented in supplemental figure 3a of that paper: <https://doi.org/10.1038/nmeth.4306>

The authors should refer to this work as it corroborates their approach.

Point 3:

Bar graphs are good for counts but not for displaying average values. This is especially true when the axis doesn't start at zero, since in these cases the length of the bar is not proportional to the value it reflects. Therefore, the bars need to be discarded for the plots in figure panels 1e, 1h, 1k, 2e.

Point 4:

I encourage the authors to show all data in figure panels that only show bars as summary (e.g. 2g-i, 3e-g, 4b-d, 4h-j, 5b-d, all plots in figure 6, 7f-h). Ideally superplots are used to distinguish technical from biological replicates: <https://doi.org/10.1083/jcb.202001064>

Point 5:

The sensor output is compared for two compartments, i.e. cytosol and plastid (figures 2a-2e, 7a-7d). For this quantitative comparison, it is assumed that (i) the dynamic range is identical and (ii) the affinity is identical. Please mention these assumption in the text to inform the reader.

The assumption that a sensor has the same dynamic range in different compartments is not necessarily true. For a good example, see figure 1 of: <https://www.nature.com/articles/ncomms15031>

Since the observed ratios between the compartments are strikingly different (e.g. in panel 1b and 7b, 7d), I wonder to what extent the assumptions are met?

Since two fluorescent proteins are used (CFP/YFP for the ATP sensors and cpYFP and mCherry for the NAD(P)H sensors, it is possible that the maturation is different for the different compartments (or due to other factors, i.e. chloride concentration or ionic strength).

As such, establishing the dynamic range in the two different compartments is important. (I realize that determining the affinity in the different compartments is hardly possible.) The observation that the mutant ATP sensor (figure 1c) is similar is encouraging, but not sufficient as it only reports on one of the extremes of the dynamic range.

Reviewed by Joachim Goedhart (University of Amsterdam, NL).

We thank the comments of the reviewers. The following new figures/tables were added to the revised manuscript accordingly:

Fig. 5e – h, Supplementary Fig. 7 (a-f), Supplementary Fig. 8 (a-c), Supplementary Table 1 and Supplementary Table 3. The model in Fig. 8 is also amended.

Reviewer #1 (Remarks to the Author):

“Bioenergetics of pollen tube growth in *Arabidopsis thaliana* revealed by ratiometric genetically encoded biosensors” by Jinhong Liu, Shey-Li Lim, Jia Yi Zhong and Boon Leong Lim

Nature Communications Research Article

MS ID: NCOMMS-22-21551A-Z

The authors present a research article about energy generation supporting pollen tube growth in *Arabidopsis thaliana*.

The authors present an interesting manuscript and the content has been well described. In addition, the research described here is of utmost importance to understand energy generation during pollen tube development. Knowledge that is lacking so far.

In general, the authors need to improve their language (e.g. grammar). Furthermore, original research papers need to be cited and various references are lacking. Furthermore, there are several points which need to be paid attention to:

>We thank the reviewer for pointing out that original research papers should always be cited. We cited the original research paper as requested (Refs 24, 67, 68, 75, 76, 78, 79, 80, 81, 82, 83).

Regarding language improvement, the manuscript has been proofread by a professional editing company.

Minor points:

1. Lines 183-184: I guess you refer to Fig. 3 c-g, not Fig. 2 c-g.

> Thank you for pointing this out. We amended the Fig. 2 c-g to Fig. 3 c-g in line 192.

2. Lines 317-318: OAA is synthesized in the cytosol by PEP carboxylase (PEPc) 47.

What about cytosolic MDHs? These enzymes can also generate OAA in the cytosol.

>MDHs carry out the following reversible reaction:

The forward reaction has a large positive G value (+29.7 KJ/mol⁻¹) and thus favours the formation of malate. ¹⁴CO₂ experiment showed that both tulip mesophyll and epidermis mainly fixed CO₂ into malate in the dark (Willmer and Ditttrich, 1974). This is mediated by PEPc (CO₂ → OAA) and MDH (OAA → malate). Hence, it is likely that the above reactions also happen in the pollen tube.

3. Lines 402-406: Treatment with AEP or disulfiram resulted in a comparable drop in cytosolic and plastid NADH/NAD⁺ ratios, without affecting the NADPH levels in these two compartments, indicating that the NADH/NAD⁺ ratios in plastids, the cytosol, and mitochondria are interconnected 25, 44 and that NADH/NAD⁺ ratios and NADPH levels are differentially regulated (Fig. 7a–d).

In this context, malate valves and citrate circulation should be mentioned as possible routes. It is well known that plastids and mitochondria are interconnected via the cytosol. And that metabolite shuttling plays an essential role.

> Thank you for your suggestion. We added the sentence malate valves and citrate circulation also play essential roles in redox shuttle in line 438-439.

4. Lines 487-491: The PCR products for NTT1 (1.5 kb), NTT2 (1.5 kb), ALDH3I1 (0.9 kb), ALDH10A8 (1.3 kb), ALDH2B7 (1.9 kb), and ALDH7B4 (0.8 kb) promoters were then cloned upstream of the GUS gene in the binary vector pBI121::35S-GUS with the 35S promoter removed by HindIII and XmaI restriction digestion.

The promoter length of ALDH3I1 and ALDH7B4 is quite small. Why have the authors chosen such a short promoter region? In such a short promoter, various regulatory elements might be missing.

> Thank you for your question. We followed the GUS studies on ALDH3I1 and ALDH7B4 promoters when we cloned the promoters. The short promoters contain all essential regulatory elements for the genes. For ALDH3I1, we referred to (Kirch et al., 2005). For ALDH7B4, we referred to (Missihoun et al., 2011). We added the promoter information to the Supplementary Table 3.

5. Lines 516-519: Arabidopsis (*Arabidopsis thaliana*) seeds were surface-sterilized with 20% (v/v) Clorox for 15 min and washed with distilled water three times before germinating on Murashige and Skoog medium 68 supplemented with 2% (w/v) sucrose, 0.1% (w/v) 2-(N-morpholino) ethanesulfonic acid, and 0.25% (w/v) Phytigel with a pH adjusted to 5.7–5.8 with KOH.

½ MS medium? Or full-strength MS medium?

> We used full-strength MS medium for seed germination. We specified full-length Murashige and Skoog medium in line 552.

6. Fig. 3 c-g: Why haven't the authors used already published knockout lines? And what about the analysis of the double mutant? This is available.

> Actually, the NTT knockout lines we used have been published in (Lim et al., 2022). We have recently obtained the *ntt1/2* double mutant from (Reiser et al., 2004) and characterized it. We added the data to Supplementary Figure 7.

7. Discussion: It has been published that pNAD-MDH interacts with the FtsH12/FtsHi complex and Ycf2 which play an essential role in plastid development. This needs to be added to the discussion.

> Thank you for your suggestion. FtsH12 is an ATP-dependent metalloprotease. We found the interaction between pNAD-MDH and the FtsH12/FtsHi intriguing. However, we do not know how to relate this to our study. Any suggestion from the reviewer is welcome.

8. Lines 221-222: Among the three plastid pyruvate kinase genes in Arabidopsis, PKp1 was highly expressed in pollen tubes in a histochemical GUS assay 31. Hence, we conclude that both NADP-ME4 and plastid glycolysis supply pyruvate to pollen plastids.

This needs to be shown, not only concluded. Why didn't the authors analyze pollen tube growth of a *PKp1* mutant?

> There are three plastid PKp genes in the Arabidopsis genome. According to the microarray data (Supplementary table 1) and our qRT-PCR data (Supplementary Figure 8), PKp3 has the highest mRNA expression level in Arabidopsis pollen tube, suggesting that it is the major PKp that supplies pyruvate to pollen plastids during pollen tube growth. We tested the pollen tube growth of *pkp1* and *pkp2* single mutants and there were no phenotype or defects on pollen tube growth. Nonetheless, no *pkp3* mutant with T-DNA insertion into exon has been isolated so far (Lines 234-241).

9. Lines 238-249: In the PDH bypass, acetaldehyde is converted to acetate by ALDH³³. To examine whether this enzymatic step takes place within pollen plastids or in any other compartments, we evaluated the expression profiles of several Arabidopsis ALDHs in pollen tubes by reverse transcription–quantitative PCR (RT-qPCR) or GUS histochemical assays. We established that the genes encoding the only two plastid-localized ALDHs (chloroplast-localized ALDH3I1 and leucoplast-localized ALDH10A8) were not expressed in pollen tubes (Fig. 6i, j). Among other ALDHs that may participate in the acetaldehyde–acetate reaction, the genes encoding cytosolic ALDH7B4 and mitochondrial ALDH2B7 were highly expressed in growing pollen tubes, as shown by RT-qPCR and GUS staining (Fig. 6i, j). Hence, acetaldehyde is likely to be converted to acetate by ALDH2B7 in mitochondria but not in plastids. The lack of growth defects in the *acs* mutant also

indicated that the PDH bypass is not essential for Arabidopsis pollen tube growth, making the plastid PDH pathway the key source of acetyl-CoA for FA biosynthesis.

In the introduction, the authors describe that there are 16 genes encoding ALDHs but only 4 have been analyzed via GUS staining and 6 via qRT-PCR. At least, gene expression of all 16 ALDHs in pollen tubes should be analyzed.

> To date, only family 2 ALDH members (ALDH2B4, ALDH2B7, and ALDH2C4) have been shown to oxidize acetaldehyde to acetate (Ref 18, 22) and therefore we examined their expression levels by qRT-PCR (Fig. 6j and Supplementary Fig. 8). To provide more information on the expression levels of the other ALDHs to the reader, microarray data extracted from reference 60 were presented in Supplementary Table 1.

10. Lines 252-253: There are four possible sources of NADH in non-photosynthetic plastids: plastid glycolysis, pPDH, pNAD-MDH 7, and plastid ALDH 23.

That's not true. What about NADH-GOGAT, GapA/B and PGDH? These have not been considered.

> In pollens, the direction of NADH-GOGAT is to recycle NAD^+ from NADH and therefore NADH-GOGAT is unlikely to be a source of NADH (Selinski et al., 2014; Selinski and Scheibe, 2014).

GapA/B is indeed a part of cytosolic glycolysis and the knockout of GapA/B led to male sterility (Selinski and Scheibe, 2014).

We overlooked pPGDH. Thank you for pointing out this important point. PGDH is the first enzyme involved in the phosphorylated pathway in serine biosynthesis from 3-PGA by utilizing NAD^+ as cofactors. pPGDH could also be a source of PDGH in pollen plastids, as a substantial amount of pPGDH1 transcript is expressed in pollen tube (Supplementary Table 1 and supplementary Figure 8). Knocking out of pPGDH1 (EDA9) is embryo lethal (Ref. 34) and therefore we could not test its importance in pollen tube growth. The pPGDH pathway should be a major source of serine during pollen tube growth, as there is no photorespiration. Its flux is likely to be controlled by the availability of its substrate, 3PGA, the major product of Rubisco CO_2 fixation, which does not happen in pollen tube. Hence, we believe that the embryo lethal phenotype of the *ppdgh1* mutant could be due to the lack of serine supply, rather than the lack of NADH supply by pPGDH1 (Lines 284-291). We modified the model in Fig. 8 accordingly.

11. Lines 259-262: pNAD-MDH is likely to generate malate and NAD^+ instead of NADH in plastids to support plastid glycolysis and supply malate for NADP-ME4. We determined that this step is important, as a mutant (suppressor of *mod1 410 [som410]*) 7 harboring an A90V amino acid substitution in pNAD-MDH was characterized by slightly shorter pollen tubes (Fig. 7e-h).

It has already been shown in the literature that pNAD-MDH is highly expressed in pollen and pollen tubes. Furthermore, 50% of the pollen of heterozygous knockout mutants do not develop a pollen tube. However, by adding NADH-GOGAT substrates pollen tube growth could be rescued, indicating an important role of NADH-GOGAT in pollen tube growth which has not been considered in this manuscript.

>Yes. We agree that NADH-GOGAT is also important in recycling NAD^+ from NADH. We added "In knock-out mutants of *pNAD-MDH*, NADH-dependent glutamate synthase (NADH-GOGAT) can also recycle NAD^+ from NADH when its substrates were added exogenously" to lines 343-346.

12. Lines 398-400: Therefore, we conclude that the plastid PDH pathway is the key pathway to supply acetyl-CoA for FA biosynthesis during pollen tube growth.

Again, it's only a conclusion but the authors do not present any data to show that.

> The building block of FA, acetyl-CoA, can be generated from pyruvate or acetate via catalysis by plastid pyruvate dehydrogenase (pPDH) or acetyl-CoA synthetase (ACS) of the PDH bypass, respectively. As ACS of the PDH bypass is not essential for Arabidopsis pollen tube growth, it is reasonable to deduce that the plastid PDH pathway is the key pathway to supply acetyl-CoA for FA synthesis during pollen tube growth.

13. Lines 400-402: The stronger inhibitory effect on pollen tube growth of disulfiram compared with AEP in both Arabidopsis (Fig. 6b) and tobacco 32 may be due to its potential inhibition of other ALDHs.

Therefore, the authors should at least analyze transcript levels of other ALDHs (see above).

> Please refer to our reply to point 9. To provide more information on the expression levels of the other ALDHs to the reader, microarray data extracted from reference 60 were presented in Supplementary Table 1.

14. Lines 615-616: The Arabidopsis housekeeping gene EF1 α (At5g60390) was used as the internal control.

At least 2 housekeeping genes need to be used. That's standard today. One housekeeping gene is not enough.

> We thank the reviewer for pointing this out. In addition to using the EF1 α housekeeping gene shown in Figures 3b, 6j, we also included GAPDH and UBC9 as internal controls in our qRT-PCR analysis. A similar trend was observed in our results, which are presented in Supplementary Figure 8.

Reviewer #2 (Remarks to the Author):

Jinhong Liu and colleagues modify a number of biosensors and apply those to study the energy fluxes during pollen tube growth in Arabidopsis thaliana. My review focuses on the engineering, characterisation and application of the biosensors (and not on energy metabolism or plant biology) as this is my area of expertise.

The biosensor approach is suitable, but the biosensors performance needs a more thorough validation when comparing different localisations (e.g. plastids vs. cytoplasm). This issue is, together with other suggestions for improvement, further explained below.

Point 1:

The claim that 'pH insensitive' second-generation ratiometric biosensors for pyridine nucleotides were developed is an overstatement. Only a rather narrow range of pH 7.0-8.5 was tested. This statement should be downtrend accordingly.

> We thank reviewer 2 for pointing this out. We amended the sentence accordingly to a more precise sentence in lines 22 and 111. Indeed, the purified biosensor proteins were only tested between pH 7.0–8.5. We only tested this range because this is the physiological range of plant organelles we are targeting, which is chloroplast (pH 7.3) and cytoplasm (pH 7.2) (Shen et al., 2013).

Point 2:

The notion that a pH insensitive sensor for iNap can be made by fusing it to mCherry was already mentioned in the original paper: "Fusion of iNap1 and the red fluorescent protein mCherry allows ratiometric and pH-resistant measurement". And this is documented in supplemental figure 3a of that paper: <https://doi.org/10.1038/nmeth.4306>

The authors should refer to this work as it corroborates their approach.

> We agreed and added this information to lines 81-83.

Point 3:

Bar graphs are good for counts but not for displaying average values. This is especially true when the axis doesn't start at zero, since in these cases the length of the bar is not proportional to the value it reflects. Therefore, the bars need to be discarded for the plots in figure panels 1e, 1h, 1k, 2e.

> In our earlier publications and other biosensor-related publications, bar graphs have been accepted for presenting the average values of sensor ratios, some examples are shown below.

Fig. 1 Light responses of iNAP sensors in different compartments of 10-day-old cotyledon mesophyll. **a** Seedlings expressing cytosolic iNAP1, iNAP4, and iNAPc. **b** Seedlings expressing stromal iNAP1, iNAP4, and iNAPc, and **c** seedlings expressing peroxisomal iNAP1, iNAP4, and iNAPc were continuously illuminated for 180 s at $296 \mu\text{mol m}^{-2} \text{s}^{-1}$ ($n = 5$; error bars \pm SEM). Emissions at 520 nm after sequential excitation at 405 nm and 488 nm were recorded, and the ratios of the two emissions ($R_{405/408}$) are presented in pseudocolor image, where high ratios (red) correspond to high NADPH levels. Scale bars, 20 μm . Asterisks indicate statistical significance as determined using paired *t*-test: * $P < 0.05$, ** $P < 0.01$, *** $P < 0.001$.

(Lim et al., 2020)

Fig. 1 Illumination induces detectable ATP and NADPH biosynthesis in mesophyll cell chloroplasts but not in guard cell chloroplasts. The 3rd and 4th

(Lim et al., 2022)

[REDACTED]

(Behera et al., 2018)

b) Kinetics of iNap1, roGFP1 or iNapc fluorescence in HeLa cells in response to 40 μM diamide. **(c,d)** Kinetics of iNap1 (Tao et al., 2017)

As suggested by the reviewer, we changed bar graphs of the pollen tube growth data to superplots (see point 4 below).

Point 4:

I encourage the authors to show all data in figure panels that only show bars as summary (e.g. 2g-i, 3e-g, 4b-d, 4h-j, 5b-d, all plots in figure 6, 7f-h). Ideally superplots are used to distinguish technical from biological replicates: <https://doi.org/10.1083/jcb.202001064>

>This is a good suggestion. We replaced the bar graphs with superplots to distinguish technical replicates in Figs 2g-i, 3e-g, 4b-d, 4h-j, 5b-d, 5f-h, 6b-d, 6f-h 7f-h, and in Supplementary Figs 4b-d, 7d-f.

Point 5:

The sensor output is compared for two compartments, i.e. cytosol and plastid (figures 2a-2e, 7a-7d). For this quantitative comparison, it is assumed that (i) the dynamic range is identical and (ii) the affinity is identical. Please mention these assumption in the text to inform the reader.

The assumption that a sensor has the same dynamic range in different compartments is not necessarily true. For a good example, see figure 1 of: <https://www.nature.com/articles/ncomms15031>

> We agree. We reminded the readers that the biosensors may or may not have identical dynamic ranges and affinities when they are expressed in different subcellular compartments in lines 117-120.

Since the observed ratios between the compartments are strikingly different (e.g. in panel 2b and 7b, 7d), I wonder to what extent the assumptions are met?

Since two fluorescent proteins are used (CFP/YFP for the ATP sensors and cpYFP and mCherry for the NAD(P)H sensors, it is possible that the maturation is different for the different compartments (or due to other factors, i.e. chloride concentration or ionic strength). As such, establishing the dynamic range in the two different compartments is important. (I realize that determining the affinity in the different compartments is hardly possible.) The observation that the mutant ATP sensor (figure 2c) is similar is encouraging, but not sufficient as it only reports on one of the extremes of the dynamic range.

>We thank the reviewer for his understanding that determining sensor affinity in subcellular compartments is technically difficult. We have validated the ATP sensor (At1.03) in Arabidopsis mesophyll cells by carrying out experiments with various treatments (e.g. illumination, inhibitor studies, etc.). In mesophyll cells the cytosolic ATP concentration was much higher than stromal ATP concentration due to the high ATP consumption in stroma and the lack of ATP importation from cytosol (Voon et al., 2018). Here, we showed that in pollen tubes the cytosolic ATP concentration is also higher than the stromal ATP concentration (Fig. 2b). The use of an ATP non-responsive mutant sensor (Fig. 2c) as a control further supports that the difference is genuine. By inhibiting mitochondrial ATP synthesis using oligomycin, the sensor ratios of At1.03 in both stroma and cytosol dropped to a low level (Fig. 2e). The slightly higher ATP concentrations in stroma than in cytosol can be explained by ATP production via plastid glycolysis, which is not inhibited by oligomycin. By contrast, when rotenone was employed to lower mitochondrial ATP output, both stromal and cytosol ATP concentrations decreased but the cytosolic ATP concentration was still higher than the stromal ATP concentration. These data suggest that the ATP sensor behaved similarly in both compartments. Similarly, we have validated the iNAPs and SoNar sensors in Arabidopsis mesophyll cells by carrying out experiments with various treatments. In mesophyll cells the cytosolic NADPH concentration and NADH/NAD⁺ ratio were lower than stromal NADPH concentration and NADH/NAD⁺ ratio (Lim *et al.*, 2020). Here, we also showed that in pollen tubes the cytosolic NADPH concentration and NADH/NAD⁺ ratio is lower than the stromal NADPH concentration (Fig. 7d) and NADH/NAD⁺ ratio (Fig. 7b). While we cannot guarantee that the sensors have completely identical dynamic ranges and affinities in these two compartments, we believe our findings and conclusions are supported by the data.

References

Behera, S., Xu, Z.L., Luoni, L., Bonza, M.C., Doccula, F.G., De Michelis, M.I., Morris, R.J., Schwarzlander, M., and Costa, A. (2018). Cellular Ca²⁺ signals generate defined pH signatures in plants. *Plant Cell* 30, 2704-2719.

Kirch, H.H., Schlingensiepen, S., Kotchoni, S., Sunkar, R., and Bartels, D. (2005). Detailed expression analysis of selected genes of the aldehyde dehydrogenase (ALDH) gene superfamily in *Arabidopsis thaliana*. *Plant Mol Biol* *57*, 315-332. 10.1007/s11103-004-7796-6.

Lim, S.L., Flutsch, S., Liu, J., Distefano, L., Santelia, D., and Lim, B.L. (2022). Arabidopsis guard cell chloroplasts import cytosolic ATP for starch turnover and stomatal opening. *Nature Communications* *13*, 652. 10.1038/s41467-022-28263-2.

Lim, S.L., Voon, C.P., Guan, X., Yang, Y., Gardestrom, P., and Lim, B.L. (2020). *In planta* study of photosynthesis and photorespiration using NADPH and NADH/NAD⁺ fluorescent protein sensors. *Nature communications* *11* (3238), 3238.

Missihoun, T.D., Schmitz, J., Klug, R., Kirch, H.H., and Bartels, D. (2011). Betaine aldehyde dehydrogenase genes from Arabidopsis with different sub-cellular localization affect stress responses. *Planta* *233*, 369-382. 10.1007/s00425-010-1297-4.

Reiser, J., Linka, N., Lemke, L., Jeblick, W., and Neuhaus, H.E. (2004). Molecular physiological analysis of the two plastidic ATP/ADP transporters from Arabidopsis. *Plant Physiology* *136*, 3524-3536.

Selinski, J., Konig, N., Wellmeyer, B., Hanke, G.T., Linke, V., Neuhaus, H.E., and Scheibe, R. (2014). The plastid-localized NAD-dependent malate dehydrogenase is crucial for energy homeostasis in developing *Arabidopsis thaliana* seeds. *Molecular plant* *7*, 170-186.

Selinski, J., and Scheibe, R. (2014). Pollen tube growth: where does the energy come from? *Plant signaling & behavior* *9*, e977200. 10.4161/15592324.2014.977200.

Shen, J.B., Zeng, Y.L., Zhuang, X.H., Sun, L., Yao, X.Q., Pimpl, P., and Jiang, L.W. (2013). Organelle pH in the Arabidopsis endomembrane system. *Molecular plant* *6*, 1419-1437.

Tao, R.K., Zhao, Y.Z., Chu, H.Y., Wang, A.X., Zhu, J.H., Chen, X.J., Zou, Y.J., Shi, M., Liu, R.M., Su, N., et al. (2017). Genetically encoded fluorescent sensors reveal dynamic regulation of NADPH metabolism. *Nat Methods* *14*, 720-728.

Voon, C.P., Guan, X.Q., Sun, Y.Z., Sahu, A., Chan, M.N., Gardestrom, P., Wagner, S., Fuchs, P., Nietzel, T., Versaw, W.K., et al. (2018). ATP compartmentation in plastids and cytosol of *Arabidopsis thaliana* revealed by fluorescent protein sensing. *Proc. Natl. Acad. Sci. U.S.A.* *115*, E10778-E10787.

Willmer, C.M., and Dittlich, P. (1974). Carbon dioxide fixation by epidermal and mesophyll tissues of Tulipa and Commelina. *Planta* *117*, 123-132. 10.1007/BF00390794.

REVIEWER COMMENTS

Reviewer #1 (Remarks to the Author):

“Bioenergetics of pollen tube growth in *Arabidopsis thaliana* revealed by ratiometric genetically encoded biosensors” by Jinhong Liu, Shey-Li Lim, Jia Yi Zhong and Boon Leong Lim
Nature Communications Research Article

MS ID: NCOMMS-22-21551B

The authors have answered and corrected most comments proposed by the reviewers, which has greatly improved the manuscript. However, there are still many typos and a few other points (see below) that need to be paid attention to:

- Throughout the whole text: The plural of pollen is pollen, not pollens.
- There are still cases in which the authors have not cited the original literature, as for instance references 7, 44 and 45 are not the original papers.
- Page 4, line 78: levels not level
- Page 4, line 79: plastids, not plastid
- Page 4, line 79: pH insensitive; add the information that the pH range is 7.0 to 8.5
- Page 4, line 83: the cytosol
- Page 4, line 84: T-DNA insertion mutants
- Page 4, line 85: the above mentioned
- Page 6, line 128: ...were not affected by exogenous....
- Page 6, line 136: inhibits glyceraldehyde.....
- Page 7, line 175: ...to plastids or the....
- Page 7, line 178: pollen contain
- Page 8, line 192: tube assays on the homozygous....
- Page 8, line 204: ...to those of the WT....
- Page 9, line 217:supplier in plastids....
- Page 9, line 231: pollen tube growth in a
- Page 9, line 232: ...two putative bass2 null mutants...
- Page 9, line 236:....*Arabidopsis* pollen tubes suggesting...
- Page 9, line 238: pkp1, not pKp1
- Page 9, line 239: T-DNA insertion in an exon....
- Page 11, lines 280-283: We determined that this step is important, as a mutant (suppressor of mod1 410 [som410]) 7 harboring an A90V amino acid substitution in pNAD-MDH was characterized by slightly shorter pollen tubes (Fig. 7e–h).
In accordance with the fact that 50% pollen of heterozygous pNAD-MDH mutants that are haploid do not develop a pollen tube.
- Page 11, lines 285-286: Knocking out of pPGDH1 (EDA9) is embryo lethal 34 and therefore we could not test its importance in pollen tube growth.
This could be tested with the heterozygous line as it had been done with pNAD-MDH mutants.
- Page 11, line 294: pollen also contain a....
- Page 12, line 315: the genes encoding the triose phosphate....
- Page 12, line 316: the PEP/phosphate....
- Page 13, line 331: photosynthesis and the stromal ATP level is lower than the cytosolic...
- Page 14, line 371: Pollen contain many mitochondria....
- Page 14, line 389: is a key process to support....
- Page 15, line 391: insertion mutants of cytosolic enolase....
- Page 15, line 404: under non-stress conditions
- Page 16, line 427: both mutants grow normally....
- Page 16, line 439: As pollen contain many....
- Page 16, line 443: in pollen plastids and the cytosol....
- Page 16, line 444: mutant pollen grow normally....
- Page 18, line 485: in pollen tubes, the Lat52 promoter....
- Page 18, line 493: carrying the Lat52....
- Page 18, line 496: are listed in Supplementary Table 2.

- Page 19, line 511: into WT plants as mentioned above.
 - Page 19, line 518: genes used....
 - Page 19, line 532: the absorbance of 600 nm
 - Page 20, line 543: in dialysis buffer...
 - Page 20, line 544: prior to the assay..
 - Page 20, line 565: was kept at 25°C.
 - Page 33, line 910: Arrowheads in green indicate....
 - Page 35, line 926: was lower in pollen plastids than....
 - Page 41, line 966: assays in WT and homozygous nadp-me4 T-DNA insertion mutants
 - Page 43, line 980: and two homozygous bass2 T-DNA mutants
 - Page 47, line 1015: In the cytosol, F6P....
 - Page 47, line 1025: also imported into the stroma via....
 - Page 53, lines 1063-1065: in plastids and the cytosol
 - Page 53, line 1070: similar ratios in pollen of three...
 - Page 55, line 1092: confirmed with the plastid marker protein FtsZ1 fused to mRFP...
 - Page 55, line 1094: and mRFP was excited...
 - Page 58, line 1123: levels in pollen tubes after...
 - Page 58, line 1124: biological samples was used here...
 - Rebuttal:
10. Lines 252-253: There are four possible sources of NADH in non-photosynthetic plastids: plastid glycolysis, pPDH, pNAD-MDH 7, and plastid ALDH 23.
That's not true. What about NADH-GOGAT, GapA/B and PGDH? These have not been considered.
> In pollens, the direction of NADH-GOGAT is to recycle NAD⁺ from NADH and therefore NADH-GOGAT is unlikely to be a source of NADH (Selinski et al., 2014; Selinski and Scheibe, 2014).
GapA/B is indeed a part of cytosolic glycolysis and the knockout of GapA/B led to male sterility (Selinski and Scheibe, 2014).
GapA/B is Calvin-Benson Cycle enzyme and is not located in the cytosol. GapA/B is a bispecific NAD(P)-dependent GAPDH located in plastids. In the light, this enzyme uses NADP while in darkness NAD is used as a coenzyme generating NADH. In addition, it is not the knockout of GapA/B that leads to male sterility, it is the knockout of GapCp which is the plastid-localized GAPDH involved in glycolysis. This needs to be distinguished.

Reviewer #2 (Remarks to the Author):

The changes that are made by the authors have improved the manuscript. Still, there are a couple of points that need to be addressed:

Point 1:

I realise that in my first review, I made an error, for which I apologise, in pointing to panels that have bar graphs that do not start at zero. Instead of figure panels 1e, 1h, 1k, 2e, I meant figure panels 2e, 2h (which is fixed) 2k and 3e (which is fixed).
For the two remaining panels (2e and 2k) it is wrong to display the data as it is now and the argument that it was previously tolerated is unconvincing. Data plotted as a bar graph should start at zero. But, for ratiometric data, which is only qualitative (and where zero doesn't mean anything), it is recommendable to plot the data without bars (and in that case the x-axis does not need to start at zero).

Point 2:

The authors indicate that they now have used 'superplots', but I don't see it. Sure, the data is plotted as dot plots, but biological replicates are not indicated nor used for statistics as far as I can see. Please correct this and refer to the superplot reference: <https://doi.org/10.1083/jcb.202001064>
To simplify making superplots, you may have a look at this online app: <https://doi.org/10.1091/mbc.E20-09-0583>

Point 3:

I'm glad that the authors realize that "we cannot guarantee that the sensors have completely identical dynamic ranges and affinities in these two compartments ". This is also mentioned in the results section of the revised version (lines 117-120). In my opinion this is a crucial point for the interpretation of the results and it should therefore be included in the discussion as well. Clearly, this is a limitation of the FRET biosensor approach and this should be highlighted in the discussion. Also, the possible implications of this limitation should be discussed.

Reviewed by Joachim Goedhart (University of Amsterdam, NL).

REVIEWERS' COMMENTS

Reviewer #1 (Remarks to the Author):

"Bioenergetics of pollen tube growth in Arabidopsis thaliana revealed by ratiometric genetically encoded biosensors" by Jinhong Liu, Shey-Li Lim, Jia Yi Zhong and Boon Leong Lim
Nature Communications Research Article

MS ID: NCOMMS-22-21551B

The authors have answered and corrected most comments proposed by the reviewers, which has greatly improved the manuscript. However, there are still many typos and a few other points (see below) that need to be paid attention to:

- Throughout the whole text: The plural of pollen is pollen, not pollens.

> Thank you for pointing out the typos. All 'pollens' in line 73, 192, 194, 329, 935, 963, 965, 977, 1076, and 1077 are corrected.

- There are still cases in which the authors have not cited the original literature, as for instance references 7, 44 and 45 are not the original papers.

> Reference 7 is an original research paper which is suitable for citation in lines 46, 71, 83, 279 and 324. In the revised version, we added Berkemeyer et al., 1998 (now Ref. 35) to lines in lines 71 and 271. We also added Hebbelmann et al., 2012 (now Ref. 47) to line 328.

- Page 4, line 78: levels not level

> It has been corrected in line 76.

- Page 4, line 79: plastids, not plastid

> It has been corrected in line 77.

- Page 4, line 79: pH insensitive; add the information that the pH range is 7.0 to 8.5

> It has been added in line 78.

- Page 4, line 83: the cytosol

> It has been corrected in line 81.

- Page 4, line 84: T-DNA insertion mutants

> It has been corrected in line 82.

- Page 4, line 85: the above mentioned

> It has been revised in line 83.

- Page 6, line 128: ...were not affected by exogenous....

> It has been revised in line 126.

- Page 6, line 136: inhibits glyceraldehyde....

> It has been revised in line 134.

- Page 7, line 175: ...to plastids or the....

> It has been revised in line 173.

- Page 7, line 178: pollen contain

> It has been revised in line 176.

- Page 8, line 192: tube assays on the homozygous....

> It has been revised in line 190.

- Page 8, line 204: ...to those of the WT....

> It has been revised in line 202.

- Page 9, line 217:supplier in plastids....

> It has been revised in line 214.

- Page 9, line 231: pollen tube growth in a

> It has been revised in line 227.

- Page 9, line 232: ...two putative *bass2* null mutants...

> It has been revised in line 229.

- Page 9, line 236:....*Arabidopsis* pollen tubes suggesting...

> It has been revised in line 233.

- Page 9, line 238: *pkp1*, not *pKp1*

> It has been revised in line 235.

- Page 9, line 239: T-DNA insertion in an exon....

> It has been revised in line 236.

- Page 11, lines 280-283: We determined that this step is important, as a mutant (suppressor of *mod1* 410 [som410]) 7 harboring an A90V amino acid substitution in pNAD-MDH was characterized by slightly shorter pollen tubes (Fig. 7e–h).

In accordance with the fact that 50% pollen of heterozygous pNAD-MDH mutants that are haploid do not develop a pollen tube.

> Thank you for your suggestion. The previous work on the heterozygous *pnad-mdh* knockout mutant line 159 (Selinski et al., 2014) also supported that this enzyme is crucial for pollen growth. We added this to the manuscript (line 281).

- Page 11, lines 285-286: Knocking out of pPGDH1 (*EDA9*) is embryo lethal 34 and therefore we could not test its importance in pollen tube growth.

This could be tested with the heterozygous line as it had been done with pNAD-MDH mutants.

> Thank you for this very good suggestion. We amended the sentence (line 284).

- Page 11, line 294: pollen also contain a....

> It has been revised in line 293.

- Page 12, line 315: the genes encoding the triose phosphate....

> It has been revised in line 314.

- Page 12, line 316: the PEP/phosphate....

> It has been revised in line 314.

- Page 13, line 331: photosynthesis and the stromal ATP level is lower than the cytosolic...

> It has been revised in line 330.

- Page 14, line 371: Pollen contain many mitochondria....

> It has been revised in line 368.

- Page 14, line 389: is a key process to support....

> It has been revised in line 385.

- Page 15, line 391: insertion mutants of cytosolic enolase....

> It has been revised in line 385.

- Page 15, line 404: under non-stress conditions

> It has been revised in line 401.

- Page 16, line 427: both mutants grow normally....

> It has been revised in line 424.

- Page 16, line 439: As pollen contain many....

> It has been revised in line 436.

- Page 16, line 443: in pollen plastids and the cytosol....

> It has been revised in line 440.

- Page 16, line 444: mutant pollen grow normally....

> It has been revised in line 441.

- Page 18, line 485: in pollen tubes, the Lat52 promoter....

> It has been revised in line 482.

- Page 18, line 493: carrying the Lat52....

> It has been revised in line 490.

- Page 18, line 496: are listed in Supplementary Table 2.
> It has been revised in line 493.

- Page 19, line 511: into WT plants as mentioned above.
> It has been revised in line 508.

- Page 19, line 518: genes used....
> It has been revised in line 516.

- Page 19, line 532: the absorbance of 600 nm
> It has been revised in line 529.

- Page 20, line 543: in dialysis buffer...
> It has been revised in line 539.

- Page 20, line 544: prior to the assay..
> It has been revised in line 541.

- Page 20, line 565: was kept at 25°C.
> It has been revised in line 562.

- Page 33, line 910: Arrowheads in green indicate....
> It has been revised in line 921.

- Page 35, line 926: was lower in pollen plastids than....
> It has been revised in line 939.

- Page 41, line 966: assays in WT and homozygous nadp-me4 T-DNA insertion mutants
> It has been revised in line 994.

- Page 43, line 980: and two homozygous bass2 T-DNA mutants
> It has been revised in line 1015.

- Page 47, line 1015: In the cytosol, F6P....
> It has been revised in line 1067.

- Page 47, line 1025: also imported into the stroma via....
> It has been revised in line 1077.

- Page 53, lines 1063-1065: in plastids and the cytosol
> They have been revised in the supplementary figure 3.

- Page 53, line 1070: similar ratios in pollen of three...
> It has been revised in the supplementary figure 5.

- Page 55, line 1092: confirmed with the plastid marker protein FtsZ1 fused to mRFP...

> It has been revised in the supplementary figure 5.

- Page 55, line 1094: and mRFP was excited...

> It has been revised in the supplementary figure 5.

- Page 58, line 1123: levels in pollen tubes after...

> It has been revised in the supplementary figure 8.

- Page 58, line 1124: biological samples was used here...

> It has been revised in the supplementary figure 8.

- Rebuttal:

10. Lines 252-253: There are four possible sources of NADH in non-photosynthetic plastids: plastid glycolysis, pPDH, pNAD-MDH 7, and plastid ALDH 23.

That's not true. What about NADH-GOGAT, GapA/B and PGDH? These have not been considered.

> In pollens, the direction of NADH-GOGAT is to recycle NAD⁺ from NADH and therefore NADH-GOGAT is unlikely to be a source of NADH (Selinski et al., 2014; Selinski and Scheibe, 2014). GapA/B is indeed a part of cytosolic glycolysis and the knockout of GapA/B led to male sterility (Selinski and Scheibe, 2014).

GapA/B is Calvin-Benson Cycle enzyme and is not located in the cytosol. GapA/B is a bispecific NAD(P)-dependent GAPDH located in plastids. In the light, this enzyme uses NADP while in darkness NAD is used as a coenzyme generating NADH. In addition, it is not the knockout of GapA/B that leads to male sterility, it is the knockout of GapCp which is the plastid-localized GAPDH involved in glycolysis. This needs to be distinguished.

> Thank you for pointing out our mistake in messing up GapA/B and GapCp. In our previous reply, what we meant was "GapCp1/GapCp2 is indeed a part of plastid glycolysis and the knockout of *gapcp1/gapcp2* led to male sterility (Selinski and Scheibe, 2014)". GapA/B are unlikely to generate NADH in pollen plastids. GapA/B consume NAD(P)H to generate G3P in the Calvin cycle under light and their activities are inhibited in the dark via the oxidation of their regulatory cysteine residues and the formation of GapA/PRK/CP12 aggregates (Baalmann et al., 1995; Marri et al., 2005; Sparla et al., 2002).

Reviewer #2 (Remarks to the Author):

The changes that are made by the authors have improved the manuscript. Still, there are a couple of points that need to be addressed:

Point 1:

I realise that in my first review, I made an error, for which I apologise, in pointing to panels that have bar graphs that do not start at zero. Instead of figure panels 1e, 1h, 1k, 2e, I meant figure panels 2e, 2h (which is fixed) 2k and 3e (which is fixed).

For the two remaining panels (2e and 2k) it is wrong to display the data as it is now and the argument that it was previously tolerated is unconvincing. Data plotted as a bar graph should start at zero.

But, for ratiometric data, which is only qualitative (and where zero doesn't mean anything), it is recommendable to plot the data without bars (and in that case the x-axis does not need to start at zero).

> Thank you for pointing it out. We revised the two panels (2e and 2k). They are now starting from zero like panels 2b and 2c.

Point 2:

The authors indicate that they now have used 'superplots', but I don't see it. Sure, the data is plotted as dot plots, but biological replicates are not indicated nor used for statistics as far as I can see. Please correct this and refer to the superplot reference: <https://doi.org/10.1083/jcb.202001064>

To simplify making superplots, you may have a look at this online app: <https://doi.org/10.1091/mbc.E20-09-0583>

> Thank you for providing such good online tool for SuperPlots generation. We used this online app to convert the following dot plots to superplots. And now they look really good!

Figures 2g-2i, 3e-3g, 4b-4d, 4h-4j, 5b-5d, 5f-5h, 6b-6d, 6f-6h, 7f-7h, S7d-S7f

Point 3:

I'm glad that the authors realize that "we cannot guarantee that the sensors have completely identical dynamic ranges and affinities in these two compartments". This is also mentioned in the results section of the revised version (lines 117-120). In my opinion this is a crucial point for the interpretation of the results and it should therefore be included in the discussion as well. Clearly, this is a limitation of the FRET biosensor approach and this should be highlighted in the discussion. Also, the possible implications of this limitation should be discussed.

> Thank you.

Reviewed by Joachim Goedhart (University of Amsterdam, NL).

Baalmann, E., Backhausen, J.E., Rak, C., Vetter, S., and Scheibe, R. (1995). Reductive modification and nonreductive activation of purified spinach chloroplast NADP-dependent glyceraldehyde-3-phosphate dehydrogenase. *Arch Biochem Biophys* 324:201-208.

Marri, L., Trost, P., Pupillo, P., and Sparla, F. (2005). Reconstitution and properties of the recombinant glyceraldehyde-3-phosphate dehydrogenase/CP12/phosphoribulokinase supramolecular complex of *Arabidopsis*. *Plant Physiol* 139:1433-1443.

Selinski, J., Konig, N., Wellmeyer, B., Hanke, G.T., Linke, V., Neuhaus, H.E., and Scheibe, R. (2014). The plastid-localized NAD-dependent malate dehydrogenase is crucial for energy homeostasis in developing *Arabidopsis thaliana* seeds. *Molecular plant* 7:170-186.

Sparla, F., Pupillo, P., and Trost, P. (2002). The C-terminal extension of glyceraldehyde-3-phosphate dehydrogenase subunit B acts as an autoinhibitory domain regulated by thioredoxins and nicotinamide adenine dinucleotide. *J Biol Chem* 277:44946-44952.